# Differentially Private Equivalence Testing for Continuous Distributions and Applications

**Daniel Omer**
Math. Dept.
Bar-Ilan University
omerdan@biu.ac.il

**Or Sheffet**
Faculty of Engineering
Bar-Ilan University
or.sheffet@biu.ac.il

## Abstract

We present the first algorithm for testing equivalence between two continuous distributions using differential privacy (DP). Our algorithm is a private version of the algorithm of Diakonikolas et al [16]. The algorithm of [16] uses the data itself to repeatedly discretize the real line so that — when the two distributions are far apart in $\mathcal{A}_k$-norm — one of the discretized distributions exhibits large $L_2$-norm difference; and upon repeated sampling such large gap would be detected. Designing its private analogue poses two difficulties. First, our DP algorithm can *not* resample new datapoints as a change to a single datapoint may lead to a very large change in the discretization of the real line. In contrast, the (sorted) index of the discretization point changes only by 1 between neighboring instances, and so we use a novel algorithm that set the discretization points using random Bernoulli noise, resulting in only a few buckets being affected under the right coupling. Second, our algorithm, which doesn't resample data, requires we also revisit the utility analysis of the original algorithm and prove its correctness w.r.t. the original sorted data; a problem we tackle using sampling a subset of Poisson-drawn size from each discretized bin. Lastly, since any distribution can be reduced to a continuous distribution, our algorithm is successfully carried to multiple other families of distributions and thus has numerous applications.

## 1 Introduction

Differential privacy (DP), a mathematically rigorous notion that bounds the effect of any single datum on the output distribution, is the current (de facto) gold-standard of privacy preserving data analysis. By now we have a myriad of DP-algorithms for learning and for various tasks of *statistical inference*. Indeed, the design of DP-hypothesis testers is crucial for the dissipation of DP into other data-centric fields — such as economics, education and health — that analyze sensitive data in massive quantities. However, by and large the design of DP-hypothesis testing is confined to distributions over finite (and thus discrete) domains rather than contiunuous distributions.

Hypothesis testing over continuous distributions poses a special challenge due to infinitesimally small perturbations — two continuous distributions can have Total-Variation distance of $\alpha$ using, say, exponentially many intervals each with an exponentially small shift of probability mass, which clearly cannot be detected with polynomially-size sample. Luckily, this issue was resolved in the works of [10, 15, 16] who restricted the TV-distance to using $k$-intervals. Formally, these works measure distance in $\mathcal{A}_k$-*norm*: where for any two distributions $\mathcal{P}$ and $\mathcal{Q}$ we have that $\|\mathcal{P} - \mathcal{Q}\|_{\mathcal{A}_k} = \sup_{\mathcal{I}} \sum_{j=1}^{k} |\mathcal{P}(I_j) - \mathcal{Q}(I_j)|$ where $\mathcal{I}$ is a partition of the real-line $\mathbb{R}$ into $k$ intervals $I_1, I_2, ..., I_k$.

But the fact remains that continuous distributions pose a special challenge for DP-algorithm designers. In fact, releasing even a simple statistics like the median is impossible over infinite domains [6, 7].

38th Conference on Neural Information Processing Systems (NeurIPS 2024).

And yet, as we show in this work, it is possible to compare (samples from) two continuous distributions while preserving differential privacy and discern whether the two are identical or far-apart in $\mathcal{A}_k$-distance. This suggests a sharp contrast between the task of *learning* from a (single) continuous distribution and the task of *statistical inference* based on *two* continuous distributions.

**Baseline.** This sharp contrast may seem striking at first, yet on second thought, it is known that any statistical inference task that only has two possible outputs (in our case - "accept / identical" vs "reject / far-apart") can easily be made private using the Subsample-and-Aggregate framework [26] — simply run $O(1/\epsilon)$-times the non-private equivalence tester of Diakonikolas et al [16] and return the most prevalent output using simple noisy count. This gives a simple $\epsilon$-DP equivalence tester with $O((k^{4/6}\alpha^{-6/5} + k^{1/2}\alpha^{-2})/\epsilon)$ sample complexity with two clear drawbacks: (1) its large sample complexity bound and (2) its decision to accept or reject is explained as 'in the majority of the runs of the tester it decided so'. The algorithm we present in this work improves on both aspects: it has a sample complexity of $\tilde{O}\left(\max\left\{\frac{k^{4/5}}{\alpha^{6/5}}, \frac{k^{1/2}}{\alpha^2}, \frac{k^{2/3}}{\alpha\epsilon^{1/3}}, \frac{k^{1/3}}{\alpha^{4/3}\epsilon^{2/3}}, \frac{\sqrt{k}}{\alpha\epsilon}\right\}\right)$ and it is capable of producing numerical estimators directly from the data that explain its accept/reject decision.

**Our Algorithm.** The difference that makes learning substantially more difficult from equivalence testing is that when having two distributions we are capable of "pitting one against the other": roughly speaking, we can partition the real line into intervals based on points from one distribution and see whether this partition acts as a random partition for the batch of examples that come from the second distribution. This involves only *sorting* all $s$ points from our sample on the real line and dealing with the order statistics. Our algorithm is based on privatizing the equivalence tester of [16], which operates over the continuous real line. This algorithm works in two stages: In the first stage, it looks at autocorrelation statistics that involve all $(s-1)$ pairs of adjacent (post-sorting) data points. In the second stage, it equipartitions the data using into $m$ bins using a random draw of $m$ points and repeatedly runs the following operation: run a closeness $L_2$-norm based tester on the discretized $m$-bins distribution that draws $N$ new points, and should it not reject – merge every pair of adjacent bins to create a $m/2$-bins discretization and continue.

Our DP-version of this tester also works in similar stages. The first stage is almost trivially privatized (the statistics estimated in the first stage exhibits small global sensitivity) but numerous difficulties arise in the privatization of the second stage. Most notably — the fact we use datapoints to define a partition into bins. To that end, we replace it so that we use sorted-indices, implying that a bin is composed of all datapoints in sorted indices from index $\pi_i$ to index $\pi_{i+1}$ (not including). This asserts that any single bin changes by at most two datapoints between neighboring samples. Combining this with the fact that under the null-hypothesis the difference between the number of points from $\mathcal{P}$ and $\mathcal{Q}$ in each bin is proportional to the square-root of the bin size, we can bound the sensitivity in a Propose-Test-Release fashion [20] for any single bin. However, the approach of fixing sorted indices raises two concerns. The first is that we have to revisit the utility analysis of [16] because we no longer use re-sample new points to estimate the $L_2$-norm difference of the two discretized distributions but rather re-visit the same dataset from which the bin-defining datapoints are taken. We bypass this difficulty by sampling ourselves Poisson-drawn subsample of each bin, and by focusing our analysis on a single iteration that should cause us to reject.

The second concern lies in the privacy analysis, and it is the observation that a change to a single bin isn't enough to bound the sensitivity of our statistical estimator. The algorithm computes an $L_2$-approximation of the norm-difference using all bins, and in the extreme case of two neighboring datasets that differ on the very first (sorted) datapoint that ends up as the very last datapoint in the neighboring dataset we have that *all* bins shift. To that end we use the following novel idea: we add a $\text{Ber}(1/2)$-random variable to each of the bin-defining indices. On the one hand, this possible shift by 1 changes very little in terms of the analysis; on the other hand, it allows us to argue that within $\log_4(1/\delta)$ random shifts we can correlate these Bernoulli random variables so that the bins identify. Full details of our algorithm and its analysis appear in Section 3.

**Applications.** Having designed our private equivalence tester for continuous distributions under $\mathcal{A}_k$-norm, we can now apply it to test equivalence between two distributions from a family $\mathcal{C}$ of distributions which, under suitable partition, yield large enough $\mathcal{A}_k$-distance. The following fact is immediate.

**Fact 1.** *Given a univariate distribution family $\mathcal{C}$ and $\alpha$, let $k = k(\mathcal{C}, \alpha)$ be the smallest integer such that for any $f_1, f_2 \in \mathcal{C}$ it holds that $\|f_1 - f_2\|_1 \leq \|f_1 - f_2\|_{\mathcal{A}_k} + \alpha/2$. Then there exists a $(\epsilon, \delta)$-DP equivalence testing algorithm for $\mathcal{C}$ using $\tilde{O}\left(\max\left\{k^{4/5}/\alpha^{6/5}, k^{2/3}/\alpha\epsilon^{1/3}, k^{1/2}/\alpha^2, k^{1/3}/\alpha^{4/3}\epsilon^{2/3}, \sqrt{k}/\alpha\epsilon\right\}\right)$ samples.*

In Section 4 we detail, much like [16], a variety of such hypothesis-families and the sample complexities we obtain for their respective DP-equivalence testers.

## 1.1 Related Work

Over the past twenty years, significant progress has been made in distribution property testing. Initially, research focused on the sample sizes required to assess properties of arbitrary distributions with a specific support size. Goldreich and Ron [23] introduced uniformity testing, proposing an algorithm using collision statistics with a sample complexity of $\frac{\sqrt{n}}{\alpha^4}$. Batu et al [5] studied closeness testing, evaluating whether two distributions are close in total variation. Paninski [27] established the first lower bound for the sample complexity of uniformity testing at $\Omega(\frac{\sqrt{n}}{\alpha^2})$ and proposed a new uniformity test using unique element statistics. The works of [1, 10, 29] explored optimal bounds for identity testing, focusing on $\chi^2$-based testers. They established a lower bound for closeness testing at $O\left(\frac{n^{2/3}}{\alpha^{4/3}} + \frac{\sqrt{n}}{\alpha^2}\right)$, with [10] providing matching upper bounds for $L_2$ closeness testing under certain conditions. [14]. introduced a technique reducing distribution norms to $O\left(\frac{1}{n}\right)$ by expanding the domain size, demonstrating an efficient $L_2$ tester for closeness testing. Recent works [11, 15, 16] have leveraged structural assumptions for more efficient testers in various settings, including continuous cases. These studies used the $\mathcal{A}_k$ metric, aligning with the Kolmogorov distance for $k = 2$ and total variation distance for $k$ being the domain size. Goldreich [22] showed that identity tests could be reduced to uniformity testing. While earlier testers relied on proxy measures like $L_2$ and $\chi^2$, and [12, 13] demonstrated efficient sample complexity using direct $L_1$ metric testers due to low sensitivity. We refer the interested reader to Cannone's excellent survey [9].

Several recent papers have examined hypothesis testing problems under the framework of differential privacy. Cai et al [8] used $\chi^2$ statistic for identity testing, achieving a sample complexity of $\tilde{O}\left(\max\left\{\frac{\sqrt{n}}{\alpha^2}, \frac{\sqrt{n}}{\alpha^{3/2}\epsilon}, \frac{n^{1/3}}{\alpha^{5/3}\epsilon^{2/3}}\right\} \cdot \log(1/\beta)\right)$. Aliakbarpour et al [4] proposed three algorithms for differentially private uniformity and closeness testing. They privatized unique element algorithms, collision-based testers, and $\chi^2$ tests, each with specific sample complexities and limitations. Acharya et al [2] established a lower bound for private identity testing, suggesting that small expected Hamming distances might compromise privacy guarantees. They proposed a sample complexity concerning privacy and distance parameters by privatizing an $L_1$ statistic-based algorithm. Aliakbarpour et al [3] also examined closeness testing between distributions with unequal sample sizes, introducing a new technique to privatize the 'flattening' method through data permutation. Zhang [30] derived an upper bound for closeness testing by privatizing an empirical total variation method, demonstrating small sensitivity and sample complexity of $O\left(\frac{\sqrt{n}}{\alpha^2} + \frac{n^{2/3}}{\alpha^{4/3}} + \frac{\sqrt{n}}{\alpha\sqrt{\epsilon}} + \frac{n^{1/3}}{\alpha^{4/3}\epsilon^{2/3}} + \frac{1}{\alpha\epsilon}\right)$.

**Comparison to Existing Lower Bounds.** The well-known lower bound for $\mathcal{A}_k$ closeness testing in the non-private regime is given in [16]. It is equal to $O\left(\frac{k^{4/5}}{\alpha^{6/5}} + \frac{\sqrt{k}}{\alpha^2}\right)$. As far as we know, there is no known lower bound for the private regime in that task. In [2], a lower bound for the private regime in identity testing is presented as $O\left(\frac{\sqrt{n}}{\alpha^2} + \frac{n^{2/3}}{\alpha^{4/3}} + \frac{\sqrt{n}}{\alpha\sqrt{\epsilon}} + \frac{n^{1/3}}{\alpha^{4/3}\epsilon^{2/3}} + \frac{1}{\alpha\epsilon}\right)$ (where $n = k$ is the domain size), which is a simpler task than closeness testing. Additionally, [30] provided an upper bound equal to the lower bound of the closeness testing also in the private regime when he crucially relies on the $L_1$ tester that is known with low sensitivity. We have established an upper bound that is asymptotically close to the lower bound, given by $\tilde{O}\left(\max\left\{k^{4/5}/\alpha^{6/5}, k^{2/3}/\alpha\epsilon^{1/3}, k^{1/2}/\alpha^2, k^{1/3}/\alpha^{4/3}\epsilon^{2/3}, \sqrt{k}/\alpha\epsilon\right\}\right)$. The difference between our upper bound and the lower bound lies in two specific terms: $\frac{\sqrt{k}}{\alpha\epsilon}$ and $\frac{k^{2/3}}{\alpha\epsilon^{1/3}}$ The first term is a result of the high sensitivity in our algorithm due to the use of $L_2$ norm testing, which was selected to ensure the utility proof. Additionally, we did not focus on optimizing the term $\log\left(\frac{k}{\alpha\epsilon\delta}\right)$.

## 2 Preliminaries

**Equivalence (Closeness) Testing.** We assume oracle access to two distributions $\mathcal{P}, \mathcal{Q}$ which gives an i.i.d. example from the resp. distribution. We also use $\mathcal{P}(S)$ (resp. $\mathcal{Q}(S)$) to denote the total probability mass assigned by $\mathcal{P}$ (resp. $\mathcal{Q}$) to a set $S$. We assume the two distributions are continuous with no discrete point mass, which can always be obtained by concatenating each sample with a uniformly drawn number $\in_R [0, 1]$. An equivalence tester between the two distributions should return NULL w.p. $\geq 2/3$ if it holds that $\mathcal{P} = \mathcal{Q}$ and return ALT w.p. $\geq 2/3$ if it holds that $\|\mathcal{P} - \mathcal{Q}\|_{\mathcal{A}_k} \geq \alpha$.

**Differential Privacy.** Two databases $D$ and $D'$ are considered neighboring databases if they differ by exactly one record, noted as $d(D, D') = 1$, where $d(\cdot, \cdot)$ represents the Hamming distance. Given a domain $\mathcal{X}$, two multi-sets $S, S' \subset \mathcal{X}$ are called *neighbors* if they differ on a single entry. An algorithm (alternatively, mechanism) $\mathcal{M}$ is said to be $(\epsilon, \delta)$-*differentially private* (DP) [21, 19] if for any two neighboring $S, S'$ and any set $O$ of possible outputs we have: $\Pr[\mathcal{M}(S) \in O] \leq e^\epsilon \Pr[\mathcal{M}(S') \in O] + \delta$. If $\delta = 0$ then we say the algorithm $\mathcal{M}$ is $\epsilon$-DP.

The Global Sensitivity of a function $f : \mathcal{X} \to \mathbb{R}^d$ is defined as the maximal difference $\max_{S,S'} \|f(S) - f(S')\|_1$. It is known that adding $\mathrm{Lap}(GS(f)/\epsilon)$ to each coordinate of $f(S)$ is $\epsilon$-DP; where $\mathrm{Lap}(\lambda)$ denotes the Laplace Distribution with parameter $\lambda$, whose PDF is $\mathsf{PDF}(x) \propto e^{-|x|/\lambda}$. It is known [19] that if $\mathcal{A}_1$ and $\mathcal{A}_2$ are $(\epsilon_1, \delta_1)$-DP and $(\epsilon_2, \delta_2)$-DP resp., then their composition is $(\epsilon_1 + \epsilon_2, \delta_1 + \delta_2)$-DP. It is also known [18] that the $k$-fold composition of $k$ algorithms, each is $(\epsilon, \delta)$-DP is $(\epsilon^*, k\delta + \delta')$-DP for any $\delta' > 0$ and $\epsilon^* = k\epsilon(e^\epsilon - 1) + 2\epsilon\sqrt{k \log(1/\delta')}$. Lastly, it is also known that if there exists an event $\mathcal{E}$ such that under $\mathcal{E}$ holding, algorithm $\mathcal{M}$ satisfies that for any two neighboring $S$ and $S'$ and any set of outputs $O$ we have that $\Pr[\mathcal{M}(S) \in O \,|\, \mathcal{E}] \leq e^\epsilon \Pr[\mathcal{M}(S') \in O \,|\, \mathcal{E}] + \delta$ then algorithm $\mathcal{M}$ is $(\epsilon, \delta + \Pr[\neg\mathcal{E}])$-DP.

**Poisson Distribution.** The Poisson distribution $\mathrm{Poi}(\lambda)$ is a discrete distribution over the Naturals which satisfies $\Pr[k] = e^{-\lambda}\lambda^k/k!$. It has multiple properties that make it easy to work with.

**Proposition 2.**
- *The sum of two ind. Poisson $Poi(\lambda_1)$ and $Poi(\lambda_2)$ is $Poi(\lambda_1 + \lambda_2)$.*

- *Let $X_1, X_2, \ldots$ be i.i.d. Bernoulli r.v.s. with parameter $p$; then drawing $t \sim Poi(\lambda)$, it holds that $\sum_{i=1}^t X_i$ is distributed like $Poi(\lambda p)$.*

- *If $X \sim Poi(\lambda_x)$ and $Y \sim Poi(\lambda_y)$ are two ind. Poisson r.v.s, then the distribution of $X$ conditioned on the event that $X + Y = n$ is Binomial $Bin(n, \frac{\lambda_x}{\lambda_x + \lambda_y})$.*

- *If $X_i \sim Poi(\lambda)$ then $\Pr[|X - \lambda| > k] \leq 2\exp\left(-\frac{k^2}{2(\lambda + k)}\right)$. So for any $\beta > 0$, setting $k = 2\sqrt{\lambda \log(2/\beta)}$ we get $\Pr[|X - \lambda| > 2\sqrt{\lambda \log(2/\beta)}] \leq \beta$ provided $\lambda > 4\log(2/\beta)$.*

**Misc.** We use $\tilde{O}$ (resp. $\tilde{\Omega}$) to denote big-$O$ (resp. big-$\Omega$) up to poly-log factors. We made no effort to minimize constants or the degree of the poly-log.

## 3 Private Equivalence Testing for Continuous Distributions

In this section, we present our DP-tester for equivalence between two distributions with large $\mathcal{A}_k$ distance and give both its formal privacy guarantee and its sample complexity bounds. The tester is detailed in Algorithm 1 (which in turn invokes Algorithm 2), where $c_{\mathrm{dkn}}$ denotes a large enough constant detailed in [16].

The algorithm consists of two parts. In terms of the non-private algorithm, both parts function similarly to the algorithm presented by [16]. Our main contribution is the privatization of this algorithm. In the first part the main objective is to compute the estimator $Z$ (line I.5) which is privatized using a straightforward approach – since it has low sensitivity we merely add to is some Laplace noise. The second part is more complex, as it involves discretizing the domain based on the data itself, resulting in high sensitivity. We addressed this issue by creating the initial partition of the domain and adding a Bernoulli random variable to each index position (line II.5). Consequently, the algorithm can iteratively run $j_0$ iterations of the $L_2$-tester TestCloseness on based on the randomized partition, where in each invocation of TestCloseness we sample a Poisson-size batch

from each partition (line II.11). If the $L_2$-tester doesn't reject, then we merge adjacent partition cells (line II.15) and move on to the next iteration. If either invocations of TestCloseness (or the estimator $Z$ has too high of a value) then we reject; but if all test pass we return NULL.

---

**Algorithm 1** Private Equivalence Tester

---

**Input:** 2 continuous distributions $\mathcal{P}, \mathcal{Q}$, distance parameter $\alpha$, privacy parameter $\epsilon, \delta$.
**Output:** "NULL" if $\mathcal{P} = \mathcal{Q}$; "ALT" if $\|\mathcal{P} - \mathcal{Q}\|_{\mathcal{A}_k} \geq \alpha$

1: Part I:

    1. Set $m \leftarrow 100 c_{\text{dkn}} \left( k^{\frac{4}{5}} / \alpha^{\frac{6}{5}} + k^{\frac{2}{3}} / \alpha \epsilon^{\frac{1}{3}} \right)$

    2. Draw $s \sim \text{Poi}(m)$ points from $\frac{1}{2}(\mathcal{P} + \mathcal{Q})$.

    3. Label each $x_i$ drawn from $\mathcal{P}$ with $\ell(x_i) = 1$ and label each $x_j$ drawn from $\mathcal{Q}$ as $\ell(x_j) = -1$.

    4. Sort the sample, denote the outcome as $x_{(1)} < x_{(2)} < \cdots < x_{(s)}$.

    5. $Z = \sum_{i=1}^{s-1} \ell(x_{(i)}) \ell(x_{(i+1)})$

    6. Draw $X \sim \text{Lap}(6/\epsilon)$.

    7. $\tilde{Z} = Z + X$

    8. **if** $\tilde{Z} > \frac{m^3 \alpha^3}{2k^2}$ **then return** ALT

2: Part II:

    1. Set $N \leftarrow 10^7 \left( \frac{k^{1/3}}{\alpha^{4/3} \epsilon^{2/3}} + \frac{\sqrt{k}}{\alpha \epsilon} + \frac{\sqrt{k}}{\alpha^2} \right) \log^6(k/\alpha \epsilon \delta)$ and assert $m$ divides $N$.

    2. Draw $s_p \sim \text{Poi}\left(\frac{N}{2}\right)$ and $s_q \sim \text{Poi}\left(\frac{N}{2}\right)$ ind. Set $s \leftarrow s_p + s_q$.

    3. $S \leftarrow$ a sample of $s_p$ i.i.d. examples from $\mathcal{P}$ and $s_q$ i.i.d. examples from $\mathcal{Q}$.

    4. Sort $S$. Denote the outcome as $x_{(1)} < x_{(2)} \leq \cdots < x_{(s)}$.

    5. For each $1 \leq i \leq m$ set $\pi_i \leftarrow i \cdot \frac{N}{m} + B_i$ for ind. drawn $B_i \sim \text{Ber}(1/2)$. Also set $\pi_0 \leftarrow 0, \pi_{m+1} \leftarrow s + 1$.

    6. Form the Partition $\bar{\Pi}^0 = \{\Pi_1^0, \Pi_2^0, ..., \Pi_m^0\}$ where $\Pi_i^0 = \{x_{(i')} : \pi_i < i' < \pi_{i+1}\}$ for every $1 \leq i \leq m$.

    7. Set $j_0 \leftarrow 1 + \lceil \log(m/k) \rceil$ and $m^0 \leftarrow m$.

    8. **for** $j = 0, 1, \ldots, j_0 - 1$ **do**:

    9.     **for** $i = 1, 2, \ldots, m^j$ **do**:

    10.         Draw $N_i^J \sim \text{Poi}(\frac{N}{4m^j})$.

    11.         Pick a u.a.r. subset $C_i^J$ of $N_i^j$ points from $\Pi_i^j$. (If $N_i^j > |\Pi_i^j|$ then use special $\perp$ points.)

    12.         Set $X_i^j$ (resp. $Y_i^j$) as #points from $\mathcal{P}$ (resp. $\mathcal{Q}$) in $C_i^j$.

    13.     **if** TestCloseness$\left( \sum_i N_i^j, m^j, \langle X_i^j \rangle, \langle Y_i^j \rangle, \frac{\alpha}{12\sqrt{2k+1} \cdot \log(1/\alpha)}, \frac{\epsilon}{8\sqrt{j_0 \log(2/\delta)}}, \frac{\delta}{2j_0}, \delta \right) = $ ALT **then**

    14.         **return** ALT

    15.     Merge cells: Set $m^{j+1} \leftarrow \lfloor m^j/2 \rfloor$, and for each $1 \leq i \leq m^{j+1}$ set:
$$\Pi_i^{j+1} \leftarrow \Pi_{2i-1}^j \cup \{x_{(\pi_{i'})}\} \cup \Pi_{2i}^j \qquad \triangleright i' = \text{the separating index of the two bins}$$

3: **return** NULL

---

## 3.1 Privacy Proof

In this subsection, our goal is to proof the following theorem.

**Theorem 3.** *Algorithm 1 is $(2\epsilon, 2\delta)$-DP.*

The proof of Theorem 3 shows that Part I of Algorithm 1 is $\epsilon$-DP — which is very straight forward, whereas Part II of the algorithm is $(\epsilon, 2\delta)$-DP — which involves more intricate arguments. In fact, all that is required regarding the DP of Part I is the following claim. Its proof, as well as most proofs in this section, is deferred to Appendix A.

**Claim 4.** *The estimator $Z = \sum_{i=1}^s \ell(x_{(i)}) \ell(x_{(i+1)})$ in Part I of Algorithm 1 (Line 5) has global senstivity of $6$.*

**Algorithm 2** TestCloseness-$(N, m, \bar{X}, \bar{Y}, \alpha, \epsilon', \delta', \delta)$

1: Set $n_{\max} \leftarrow \max_i \{|X_i - Y_i|\}$.
2: Set $\hat{n}_{\max} \leftarrow n_{\max} + \text{Lap}(8/\epsilon)$
3: Set $Z \leftarrow \sum_i (X_i - Y_i)^2 - X_i - Y_i = \sum_i |X_i - Y_i|^2 - (X_i + Y_i)$.
4: Set $n_z \leftarrow \sqrt{\frac{6N}{m} \log(800N)} + \frac{16 \ln(1/\delta')}{\epsilon'} + 1$
5: Set $\hat{Z} \leftarrow Z + \text{Lap}(16 \log_{4/3}(2/\delta) n_z / \epsilon')$
6: **if** ($\hat{n}_{\max} < \sqrt{\frac{6N}{m} \log(800N)} + \frac{8 \ln(1/\delta')}{\epsilon'}$ and $\hat{Z} < \frac{1}{2}\alpha^2 N^2$) **then return** NULL
7: **else return** ALT

---

We thus focus for now on Part II of Algorithm 1. In Part II our output is the $(2j_0)$-long tuple $\langle \hat{n}^j_{\max}, Z^j \rangle_{j=0}^{j_0-1}$ which we may return from the $j_0$ invocations of TestCloseness. Thus, we denote $\epsilon' = \frac{\epsilon}{8\sqrt{j_0 \log(2/\delta)}}$, $\delta' = \frac{\delta}{2j_0}$ and note that each invocation of TestCloseness is with these parameters. Throughout our analysis of Part II we assume that the Poisson random variables are known to us, but leave the Bernoulli and the Laplace random variables unknown.

Each invocation of TestCloseness releases two statistics: an approximation of $n_{\max}$ and approximation of $Z$. The next two claims bounds their sensitivity (w.h.p.).

**Claim 5.** *The Global Sensitivity of $n_{\max}$ is at most* $4$.

**Lemma 6.** *There exists an event $\mathcal{E}$ where $\Pr[\neg \mathcal{E}] < 3\delta/2$, and under $\mathcal{E}$ it holds that the sensitivity of $Z$ at any iteration $j$ is at most* $8 \log_{4/3}(2/\delta) \left( \sqrt{\frac{6N}{m} \log(800)} + \frac{16 \ln(1/\delta')}{\epsilon'} + 1 \right)$.

*Proof.* Denote $\mathcal{E}_0$ as the event that there exists an invocation of TestCloseness with $m$ balls and $n$ bins in which $n_{\max} \geq \sqrt{\frac{6m}{n} \log(800m)} + \frac{16 \ln(1/\delta')}{\epsilon'}$, yet $\hat{n}_{\max} < \sqrt{\frac{6m}{n} \log(800m)} + \frac{8 \ln(1/\delta')}{\epsilon'}$. It follows that there must exists an invocation of TestCloseness in which the noise added to $n_{\max}$ (drawn from $\text{Lap}(8/\epsilon')$) must be smaller than $-\frac{8 \ln(1/\delta')}{\epsilon'}$. This holds w.p. $< \delta'/2$, and from the Union Bound it follows that $\Pr[\mathcal{E}_0] = \frac{j_0 \cdot \delta'}{2} = \frac{\delta}{2}$.

We now turn to the Bernoulli random variables. Fix $S$ and $S'$ to be two neighboring inputs, and again, we assume that the changed point appears in place $(1)$ in $S$ and in place $(s)$ in $S'$. (Otherwise, our analysis is only simpler.) It follows that the changed point falls in $\Pi_1^0$ in $S$ and in $\Pi_m^0$ in $S'$.

We now specify the coupling of the Bernoulli random variables between $S$ and $S'$, which is the following. We draw $B_1$, $B_1'$. $B_2$, $B_2'$ and so on, u.a.r. and independently, and apply $B_i$-s to the invocation on $S$ and $B_i'$-s to the invocation on $S'$, *until the first occurrence of $B_i = 0, B_i' = 1$* from which we give $B_{i+1}$, $B_{i+2}$ and so on to both invocation. This is depicted in Figure 1.

Denote $\mathcal{E}_j$ as the event that the first occurrence of $B_i = 0, B_i' = 1$ is at the $j$-th draw. Clearly $\Pr[\mathcal{E}_j] = (3/4)^{j-1} \cdot 1/4$. Thus, from the disjointness of events, it follows that $\Pr\left[\bigcup_{j=1}^{\log_{4/3}(2/\delta)} \mathcal{E}_j\right] = \frac{1}{4} \sum_{j=1}^{\log_{4/3}(2/\delta)} (3/4)^{j-1} = \frac{1}{4} \frac{1 - (3/4)^{\log_{4/3}(2/\delta)}}{1 - 3/4} = 1 - \delta/2$. Symmetrically, we apply the same coupling

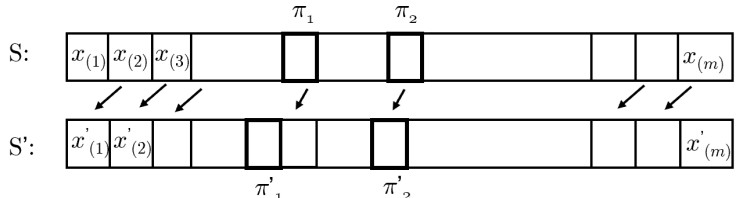

Figure 1: Two neighboring inputs that differ on one datapoint, appearing first in $S$ and last in $S'$. In this example, the index defining the first bin, $\pi_1$, is such that for $S$ we go $B_1 = 0$ and for $S'$ we have $B_1' = 0$; but the index defining the 2nd bin, $\pi_2$ it does hold that $B_2 = 0, B_2' = 1$ so the indices starting from bin 2 onwards align.

to align the last bins, those affected by $x_{(s)}$, in the last position. We use the coupling that provides $B_m, B'_m, B_{m-1}, B'_{m_1}, B_{m-2}, B'_{m_2}$ and so on until the first occurrence of $(1,0)$ then it switches to the same variable $B_j$. Denoting $\mathcal{E}'_j$ as the event that the first occurrence of $B_i = 1, B'_i = 0$ is at the $m-j$-th draw, we have that $\Pr\left[\bigcup_{j=1}^{\log_{4/3}(2/\delta)} \mathcal{E}'_j\right] = 1 - \delta/2$.

We thus denote $\mathcal{E} = \mathcal{E}_0 \cup \bigcup_{j=1}^{\log_{4/3}(2/\delta)} (\mathcal{E}_j \cup \mathcal{E}'_j)$ and have that $\Pr[\neg\mathcal{E}] = 3\delta/2$. Note, under $\mathcal{E}$ it follows that at most $2\log_{4/3}(2/\delta)$ bins have a change in their $|X_i - Y_i|$-value, which – due to Claim 5 – is at most 4. (Observe that in the case where $S$ and $S'$ are such that there are fewer than $2\log_{4/3}(2/\delta)$ bins between the locations of the changed point then this statement holds w.p. 1.)

It follows that under $\mathcal{E}$ we have that the value of $Z$ is affected by at most $2\log_{4/3}(2/\delta)$ bins, where for each bin $|X_i - Y_i|$ changes by at most 4. It follows that $Z$ can change by at most
$$2\log_{4/3}(2/\delta) \cdot 4 \cdot (n_{\max} + 1) \le 8\log_{4/3}(2/\delta)\left(\sqrt{\frac{6N}{m}\log(800N)} + \frac{16\ln(1/\delta')}{\epsilon'} + 1\right). \qquad \square$$

Completing the Proof of Theorem 3 is simple, and is deferred to Appendix A.

## 3.2 Utility Proof

In this subsection, our goal is to proof the following theorems.

**Theorem 7.** *W.p.* $\ge 2/3$, *Algorithm 1 returns* NULL *when* $\mathcal{P} = \mathcal{Q}$.

**Theorem 8.** *W.p.* $\ge 2/3$, *Algorithm 1 returns* ALT *when* $\|\mathcal{P} - \mathcal{Q}\|_{\mathcal{A}_k} \ge \alpha$.

The proof of Theorem 7 is fairly simple, but the proof of Theorem 8 requires some preliminaries. In fact, in order to argue the correctness of theorems we need to argue that both Parts I & II of the algorithm are correct w.p. $\ge 5/6$. Proving that Part I is correct is very straight-forward due to claims from [16]. And so we focus on the correctness of Part II. Its correctness requires that we first assert a few rudimentary propositions and claims.

### 3.2.1 Rudimentary Claims

Similarly to the analysis in [16] our goal is to compare the two continuous distributions to their resp. discretizations that were forms under the various $\bar{\Pi}^j$. However, we do not have the luxury of resampling the points from $\mathcal{P}$ and $\mathcal{Q}$, and so our argument diverges from theirs as we fix one particular $j^*$ and then argue from first principles that under large $\mathcal{A}_k$-difference the specific partition $\bar{\Pi}^{j^*}$ cause us to reject. (Arguing that when $\mathcal{P} = \mathcal{Q}$ we return null is also fairly straight-forward.) Our intermediate goal is to argue that the points from $\mathcal{P}$ (the $X$-s) and the points from $\mathcal{Q}$ (the $Y$-s) are distributed like independent Poisson random variables. Thus, for the remainder of the discussion we fix some particular iteration $j$ and examine *solely* it, without considering the previous iterations.

Due to space constraint, we defer nearly all the claims in this section to Appendix B, but they all lead to towards the following main lemma.

**Lemma 9.** *Fix index* $j$. *For each index* $i$ *denote by* $I_i$ *as the interval* $(x_{(\pi^j_i)}, x_{(\pi^j_{i+1})})$ *which is the interval defining bin* $i$ *in the partition* $\bar{\Pi}^j$. *Then the estimator* $Z$ *computed by* TestCloseness *satisfies that* $\mathbb{E}[Z] = \sum_{i=1}^{m^j}(p_i - q_i)^2$ *where*
$$p_i = \frac{N\mathcal{P}(I_i)}{4m(\mathcal{P}(I_i) + \mathcal{Q}(I_i))}, \qquad q_i = \frac{N\mathcal{Q}(I_i)}{4m(\mathcal{P}(I_i) + \mathcal{Q}(I_i))}.$$
*and that* $Var[Z] = \sum_{i=1}^{m^j} 4(p_i + q_i)(p_i - q_i)^2 + 2(p_i + q_i)^2$.

In Lemma 9 we established the expectation and variance of our estimator using the notation $p_i$ which in turn is defined as $\frac{N\mathcal{P}(I)}{4m(\mathcal{P}(I)+\mathcal{Q}(I))}$ (and $q_i$ similarly). The following claim is going to assist us in bounding the denominators of $p_i$ and $q_i$.

**Claim 10.** *If* $N > 3000m\log(2m)$, *then with a probability of* $1 - \frac{1}{m}$, *any* $I$ *that forms a bin* $\Pi^0_i$ *in* $\bar{\Pi}^0$, *has that* $\frac{1}{1.01m} \le \frac{\mathcal{P}(I)+\mathcal{Q}(I)}{2} \le \frac{1.01}{m}$.

Unfortunately, due to space constraints, we defer proof of Claim 10 as well as *the entire proof of Theorem 7* to Appendix B.

### 3.2.2 Proof of Theorem 8

We now turn our attention to proving Theorem 8, however, we need one more technical lemma, whose proof – like all proofs in this section – is deferred to Appendix B. Then we can proof Theorem 8.

**Lemma 11.** *Fix $p \in (0,1)$. Suppose there exists $n$ non-negative random variables $X_1, X_2, \ldots, X_n$, such that for each $i$ it holds that for some fixed number $a_i$ we have $\Pr[X_i \geq a_i] \geq p$. Then, given a constant $c > 0$ there exists another constant $C > 0$ such that with a probability at least $C$ it holds that $\sum_{i=1}^{n} X_i \geq c \sum_{i=1}^{n} a_i$.*

*Proof of Theorem 8.* In the alternative case, where $\|\mathcal{P} - \mathcal{Q}\|_{\mathcal{A}_k} \geq \alpha$, we know that there exists $k$ intervals $I \in \mathcal{I}$ such that $\sum_{I \in \mathcal{I}} |\mathcal{P}(I) - \mathcal{Q}(I)| \geq \alpha$. We denote for each interval $\gamma(I) = |\mathcal{P}(I) - \mathcal{Q}(I)|$. For the sake of analyze we use two different kinds of interval, small and large.

**Definition 12.** *Interval $I \in \mathcal{I}$ is called* small *if there exists a subinterval $J \subseteq I$ such that $\mathcal{P}(J) + \mathcal{Q}(J) < 1/m$ and $|\mathcal{P}(J) - \mathcal{Q}(J)| \geq \gamma(I)/10$, and* large *otherwise.*

Note that $\alpha \leq \|\mathcal{P} - \mathcal{Q}\|_{\mathcal{A}_k} = \sum_{I \in \mathcal{I}} |\mathcal{P}(I) - \mathcal{Q}(I)| = \sum_{I \in \mathcal{I},\, I \text{ small}} \gamma(I) + \sum_{I \in \mathcal{I},\, I \text{ large}} \gamma(I)$, so half of the discrepancy comes from either small or large intervals. We consider first the case where half of the discrepancy comes from small intervals. In this case, Lemma 9 in [14] states that the expectation of statistic $Z$ in Line 5 of Part I is bounded by $\mathbb{E}[Z] \geq c \frac{N^3 \alpha^3}{k^2}$ for some constant $c > 0$. Just like we did in the soundness case, Proposition 21 gives that the variance $\mathrm{Var}[Z] \leq 9m$. Therefore for some large constant $c_{\mathrm{dkn}}$, setting $m = 100 c_{\mathrm{dkn}} \left( \frac{k^{\frac{4}{5}}}{\alpha^{\frac{6}{5}}} + \frac{k^{\frac{2}{3}}}{\alpha \epsilon^{\frac{1}{3}}} \right)$ then by Chebyshev's inequality we get that with probability $\frac{5}{6}$ it holds that $\tilde{Z} \geq \mathbb{E}[Z] - 10\sqrt{m} - |X| \geq \frac{m^3 \alpha^3}{2k^2}$ (with $X$ being the random noise sampled in Line 6 of Part I of the algorithm).

Now consider the case where $\sum_{I \in \mathcal{I},\, I \text{ is large}} |\mathcal{P}(I) - \mathcal{Q}(I)| \geq \frac{\alpha}{2}$. We prove that Part II of Algorithm 1 must return ALT on at least one invocation of TestCloseness. In order to do this, we first need to show that the discretization of the domain with $m$ samples preserves most of the $\mathcal{A}_k$-distance. Take any interval $I \in \mathcal{I}$ that gives the $\mathcal{A}_k$ distance, and denote $I = [a, b]$ as its boundaries.

Now, consider the total probability mass between $a$ and $\pi_i$, then the first datapoint selected for the partition that is greater than $a$. Since the total number of points is taken from a Poisson distribution, it is known that the mass from one point to the next has Exponential distribution $\mathrm{Exp}(N)$ [28]; and so the total mass from $a$ to $\pi$ is distributed like the sum of Exponential random variables, namely, a Gamma-distribution with mean $\leq \frac{N}{m} \cdot \frac{1}{N} = \frac{1}{m}$, and variance $\leq \frac{N}{m} \cdot (\frac{1}{N})^2 = \frac{1}{mN} < \frac{1}{3000 m^2 \log(m)}$. It follows that w.p. $\leq 0.01$ we have that $\frac{1}{2}(\mathcal{P}[a, \pi_i] + \mathcal{Q}[a, \pi_i]) > \frac{1.01}{m}$. A similar analysis shows that for $\pi_j$, the last datapoint selected for the partition before $b$, we also have $\mathcal{P}([\pi_j, b]) + \mathcal{Q}([\pi_j, b]) < \frac{2.02}{m}$.[1] Note that a large interval must have a total probability mass $\mathcal{P}(I) + \mathcal{Q}(I) \geq \frac{10}{m}$ and so w.p. $\geq 0.98$ we have formed the subinterval $I' = [\pi_i, \pi_j] \subset I$. Moreover, by the subinterval property of large intervals, we have that because $\mathcal{P}([a, \pi_i]) + \mathcal{Q}([a, \pi_i]) + \mathcal{P}([\pi_j, b]) + \mathcal{Q}([\pi_j, b]) < \frac{4.04}{m}$ then $\gamma(I') \geq \gamma(I) - \frac{5\gamma(I)}{10} = \gamma(I)/2$.

Therefore, denote $N_I$ as the indicator of the event that $|\mathcal{P}(I') - \mathcal{Q}(I')| \geq \frac{\gamma(I)}{2}$, and we denote $\mathcal{P}^{\Pi^0}$ and $\mathcal{Q}^{\Pi^0}$ as the reduced discretized distribution formed by the partition point. We show it preserves most of the total variation $\|\mathcal{P}^{\Pi^0} - \mathcal{Q}^{\Pi^0}\|_1 = \sum_{I'} |\mathcal{P}(I') - \mathcal{Q}(I')| \geq \sum_{I \in \mathcal{I}} N_I \frac{\gamma(I)}{2} \geq \sum_{I \in \mathcal{I},\, I \text{ large}} N_I \frac{\gamma(I)}{2}$. We have established that if $I$ is large interval then $\Pr[N_I] \geq 0.98$, and so, for the random variables, $\left( N_I \frac{\gamma(I)}{2} \right)$-s (one for each large interval) where for each variable with probability $0.98$ it holds that $N_I \frac{\gamma(I)}{2} \geq \frac{\gamma(I)}{2}$, we apply Lemma 11 with $c = 0.36$ and have that there for the constant

---

[1] if $a = -\infty$ or $b = \infty$ our analysis is even simpler, as we take $\pi_0$ and $\pi_{m+1}$ as the respective partition point.

$C = 1 - \frac{0.98(1-0.98)}{0.98^2(1-0.36)^2} > 0.95$ such that with probability $C > 0.95$

$$\sum_{I \in \mathcal{I}, \, I \text{ large}} N_I \frac{\gamma(I)}{2} \geq 0.36 \sum_{I \in \mathcal{I}, \, I \text{ large}} \frac{\gamma(I)}{2} > \frac{\alpha}{12}$$

Therefore, with probability $> 0.95$, $\|\mathcal{P}^{\Pi^0} - \mathcal{Q}^{\Pi^0}\|_1 \geq \frac{\alpha}{12}$.

Observe that the TV-distance follows from finding suitable subintervals $I'$ inside large intervals with large discrepancy. Thus, each of the $\leq k$ large intervals now gives (at most) 2 points that form $I'$, so these $\leq 2k$ points partition the real line to $\leq 2k+1$ intervals where the various $I'$-s are part of this partition. It follows that $\|\mathcal{P}^{\Pi^0} - \mathcal{Q}^{\Pi^0}\|_{\mathcal{A}_{2k+1}} \geq \frac{\alpha}{12}$.

Now, to complete the proof, we need to show that Part I of the algorithm detects the discrepancy. We deploy the following lemma from [16] where for a vector $v$ the notation $\|v\|_{1,k}$ refers to the sum of the largest $k$ bins/coordinates of $v$.

**Lemma 13.** *For any two distributions $\mathcal{P}$ and $\mathcal{Q}$ on $[m]$ such that $\|\mathcal{P} - \mathcal{Q}\|_{\mathcal{A}_k} > \alpha$, there iteration $j \in [\log(m/k)]$ such that $\|\mathcal{P}^{\Pi^j} - \mathcal{Q}^{\Pi^j}\|_{1,k} > \alpha/\log(m/k)$.*

So since $\|\mathcal{P}^{\Pi^0} - \mathcal{Q}^{\Pi^0}\|_{\mathcal{A}_{2k+1}} \geq \frac{\alpha}{12}$, we know that by Lemma 13, there exists some $j^* \in [\log(m/k)]$ such that $\left\|\mathcal{P}^{\Pi^{j^*}} - \mathcal{Q}^{\Pi^{j^*}}\right\|_{1,2k+1} \geq \frac{\alpha}{12\log(m/k)}$ and therefore by Cauchy–Schwarz inequality $\left\|\mathcal{P}^{\Pi^{j^*}} - \mathcal{Q}^{\Pi^{j^*}}\right\|_2 \geq \frac{\alpha}{12\sqrt{2k+1}\log(m/k)}$. We know that $\left\|\mathcal{P}^{\Pi^{j^*}} - \mathcal{Q}^{\Pi^{j^*}}\right\|_{1,2k+1} \geq \frac{\alpha}{12\log(m/k)}$. Denote the set of indices of these $2k+1$ intervals as $\mathcal{S}$, we get from Lemma 9 that

$$\mathbb{E}[Z] = \sum_{i=1}^{m^j} (p_i - q_i)^2 \geq \sum_{i' \in \mathcal{S}} (p_{i'} - q_{i'})^2 \overset{\text{Cau.Sch.}}{\geq} \frac{\left(\sum_{i' \in \mathcal{S}} |p_{i'} - q_{i'}|\right)^2}{2k+1} \geq \frac{1}{2k+1} \left(\sum_{i' \in \mathcal{S}} \frac{N|\mathcal{P}(I) - \mathcal{Q}(I)|}{4m^j|\mathcal{P}(I) + \mathcal{Q}(I)|}\right)^2.$$

Also from Lemma 9 we can infer that

$$\Pr\left[|Z - \mathbb{E}[Z]| \leq \frac{1}{4}\mathbb{E}[Z]\right] \leq \frac{16\text{Var}[Z]}{\mathbb{E}[Z]^2} \leq \frac{\sum_{i=1}^{m^j} 64(p_i + q_i)(p_i - q_i)^2 + 32(p_i + q_i)^2}{(\sum_{i=1}^{m^j} (p_i - q_i)^2)^2}$$

$$= \frac{\sum_{i=1}^{m^j} 64(\frac{N(\mathcal{P}(I) + \mathcal{Q}(I))}{4m^j(\mathcal{P}(I) + \mathcal{Q}(I))})(p_i - q_i)^2 + 32(\frac{N(\mathcal{P}(I) + \mathcal{Q}(I))}{4m^j(\mathcal{P}(I) + \mathcal{Q}(I))})^2}{(\sum_{i=1}^{m^j} (p_i - q_i)^2)^2} = \frac{2N^2/m^j}{(\sum_{i=1}^{m^j} (p_i - q_i)^2)^2} + \frac{16N/m^j}{\sum_{i=1}^{m^j} (p_i - q_i)^2}$$

$$\overset{m^j \geq k}{\leq} \frac{6 \cdot 588^2 (2k+1) \log^2(m/k)}{N^2 \alpha^4} + \frac{28224 \log(m/k)}{N\alpha^2} < 0.01$$

when $N \geq 10^6 \frac{\sqrt{k}\log^2(m/k)}{\alpha^2}$. Now we assert that the noise we added is also proportional to at most $\frac{1}{4}\mathbb{E}[Z]$, and indeed if we draw a random variable $R \sim \text{Lap}\left(\frac{\Delta(Z)}{\epsilon'}\right)$ with $\Delta(Z)$ as defined in Line 5 of TestCloseness, then we get $\Pr\left[R \geq \frac{N^2\alpha^2}{2500(2k+1)\log(m/k)}\right] \leq \exp\left(\frac{N^2\alpha^2\epsilon'}{2500(2k+1)\Delta(Z)}\right) \leq 0.01$. which holds if

$$N > 10^6 \frac{\sqrt{k\Delta(Z)}}{\alpha\sqrt{\epsilon'}} \geq \frac{10^6}{\alpha\sqrt{\epsilon'}} \sqrt{k \cdot 16 \log_{4/3}(2/\delta) \max\left\{\sqrt{\frac{N}{2k}\log(10N)}, \frac{16\ln(1/\delta')}{\epsilon'}\right\}}$$

so we get $N = \tilde{\Omega}\left(\frac{k^{1/3}}{\alpha^{4/3}\epsilon^{2/3}} + \frac{\sqrt{k}}{\alpha\epsilon}\right)$. Combining it all together, we have that w.p. $\geq 5/6$ Part II of the algorithm also returns ALT. $\square$

# 4 Applications

Our algorithm is designed for continuous distributions, but it can also be used for discrete distributions. The process is simple: for a given discrete distribution $\mathcal{P}$, for each example $x_j \sim \mathcal{P}$ we draw a number $i_j \in_R [0,1]$, and then sort all the examples $\langle (x_j, i_j) \rangle_{j=1}^m$ using lexicographic order. This process gives a simple privatization of the "Flattening method" proposed by Diakonikolas et al [14],

as it does so without looking at the example drawn. Our algorithm method is quite simple: draw $m$ samples from $\frac{1}{2}(\mathcal{P} + \mathcal{Q})$, then calculate the autocorrelation to identify discrepancies within small intervals. Next, use $2\min(m, n)$ for flatting to test for total variation distance. It's important to remember that if $n = k$, where $n$ is the size of the domain, then $\mathcal{A}_k$ distance is equal to $L_1$. However, the sample complexity of the first part, which involves finding discrepancies in small intervals using the analysis of [16], requires $\frac{n^{4/5}}{\alpha^{6/5}}$. Fact 1 indicates however that the size of the domain is not always the most suitable parameter for distribution testing. Having knowledge about the structure of the distribution enables more efficient testing, as we illustrate below. In Table 1, we give a brief summary of the various statistical inference tasks that can be conducted using our algorithm.

Table 1: Private equivalence testers derived from our algorithm for continuous distributions

| Distrib. Family | Num of Intervals | Private upper bound |
|---|---|---|
| $t$-piecewise constant | $t$ | $\tilde{O}\left(\max\left\{\frac{t^{4/5}}{\alpha^{6/5}}, \frac{t^{2/3}}{\alpha\epsilon^{1/3}}, \frac{t^{1/2}}{\alpha^2}, \frac{t^{1/3}}{\alpha^{4/3}\epsilon^{2/3}}, \frac{\sqrt{t}}{\alpha\epsilon}\right\}\right)$ |
| $t$-piecewise degree-$d$ | $t(d+1)$ | $\tilde{O}\left(\max\left\{\frac{(t(d+1))^{4/5}}{\alpha^{6/5}}, \frac{(t(d+1))^{2/3}}{\alpha\epsilon^{1/3}}, \frac{(t(d+1))^{1/2}}{\alpha^2}, \frac{(t(d+1))^{1/3}}{\alpha^{4/3}\epsilon^{2/3}}, \frac{\sqrt{(t(d+1))}}{\alpha\epsilon}\right\}\right)$ |
| log-concave | $\frac{1}{\sqrt{\alpha}}$ | $\tilde{O}\left(\max\left\{\frac{1}{\alpha^{9/5}}, \frac{1}{\alpha^{4/3}\epsilon^{1/3}}, \frac{1}{\alpha^3\epsilon^{2/3}}, \frac{1}{\alpha^{5/4}\epsilon}\right\}\right)$ |
| $k$-mixture of log-concave | $\frac{k}{\sqrt{\alpha}}$ | $\tilde{O}\left(\max\left\{\frac{t^{4/5}}{\alpha^{8/5}}, \frac{k^{2/3}}{\alpha^{4/3}\epsilon^{1/3}}, \frac{k^{1/2}}{\alpha^{9/5}}, \frac{k^{1/3}}{\alpha^3\epsilon^{2/3}}, \frac{\sqrt{k}}{\alpha^{5/4}\epsilon}\right\}\right)$ |
| $t$-model over $[n]$ | $\frac{t\log(n)}{\alpha}$ | $\tilde{O}\left(\max\left\{\frac{(t\log(n))^{4/5}}{\alpha^2}, \frac{(t\log(n))^{2/3}}{\alpha^{5/2}\epsilon^{1/3}}, \frac{(t\log(n))^{1/2}}{\alpha^{5/2}}, \frac{(t\log(n))^{1/3}}{\alpha^{5/3}\epsilon^{2/3}}, \frac{\sqrt{t\log(n)}}{\alpha^{3/2}\epsilon}\right\}\right)$ |
| MHR over $[n]$ | $\frac{\log(n/\alpha)}{\alpha}$ | $\tilde{O}\left(\max\left\{\frac{(\log(n/\alpha))^{4/5}}{\alpha^2}, \frac{\log(n/\alpha)^{2/3}}{\alpha^{5/3}\epsilon^{1/3}}, \frac{(\log(n/\alpha))^{1/2}}{\alpha^{5/2}}, \frac{(\log(n/\alpha))^{1/3}}{\alpha^{5/3}\epsilon^{2/3}}, \frac{\sqrt{\log(n/\alpha)}}{\alpha^{3/2}\epsilon}\right\}\right)$ |

## Acknowledgments and Disclosure of Funding

O.S. is supported by the BIU Center for Research in Applied Cryptography and Cyber Security in conjunction with the Israel National Cyber Bureau in the Prime Minister's Office, and by ISF grant no. 2559/20. Both authors thank the anonymous reviewers for their suggestions and advice on improving this paper.

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

## A Missing Proofs — Privacy

**Claim 14** (Claim 4 restated). *The estimator $Z = \sum_{i=1}^{s} \ell(x_{(i)})\ell(x_{(i+1)})$ in Part I of Algorithm 1 (Line 5) has global senstivity of $6$.*

*Proof.* Consider omitting a single datapoint, $x_{(i)}$. It is easy to check cases and see that when $\ell(x_{(i-1)}) = \ell(x_{(i+1)}) = -\ell(x_{(i)})$ then before omitting $x_{(i)}$ the three entries contributed $-2$ to $Z$ andafter the omittance they contribute $1$ to $Z$. Thus the change is $3$. Considering $S$ and $S'$ that are different on a single entry, we have that this change to a triplet of consecutive datapoints occurs twice, hence the global senstivity of $Z$ is $2 \cdot 3 = 6$. $\qquad\square$

**Claim 15** (Claim 5 restated). *The Global Sensitivity of $n_{\max}$ is at most $4$.*

*Proof.* Fix $S$ and $S'$ to be two neighboring inputs that differ on a single point, and assume that the changed point appears in first place in $S$ and the last ($s$-th) place in $S'$. Now, it is simple to see that each bin changes by at most two points (its first and last), as shown in Figure 1. However, since we may not know the beginning-points and end-points of each bin due to the Bernoulli random noise added to each, then in the extreme case the bin from $S$ has addition more point and from $S'$ missed a point. This sums to $4$ points that may belong to one bin and not the other, shifting the value of $X_i - Y_i$ by at most $4$. $\qquad\square$

*Proof of Theorem 3.* First, we discuss Part I. In this part we only release the noisy estimator $\left(\sum_{i=1}^{s} \ell(x_{(i)})\ell(x_{(i+1)})\right) + X$. Due to Claim 4 we know that adding $\mathrm{Lap}(6/\epsilon)$ to this estimator is $\epsilon$-DP. We now move to Part II of the algorithm.

In this part we release $j_0$-times a pair of outputs: (i) a noisy estimator of $n_{\max}$ and (ii) a noisy estimator of $Z$. Assume the event from Lemma 6 holds. Adding $Lap(8/\epsilon')$-noise to $n_{\max}$ which has global sensitivity of $4$ (Claim 5) is $\epsilon'/2$-DP. Adding noise of $\mathrm{Lap}\left(16\log_{4/3}(2/\delta)\left(\sqrt{\frac{6m}{n}}\log\left(800m\right) + \frac{16\ln(1/\delta')}{\epsilon'} + 1\right)/\epsilon'\right)$ is $\epsilon'/2$-DP. It follows that under $\mathcal{E}$, each pair of outputs preserves $\epsilon'$-DP. Note that we set $\epsilon' = \frac{\epsilon}{8\sqrt{j_0\log(2/\delta)}}$, so it follows from Advanced Composition, that overall we preserve $(\epsilon, \delta/2)$-DP under $\mathcal{E}$.

Summing both Parts together, Algorithm 1 preserves $(2\epsilon, \delta/2)$-DP under an event that holds w.p. $1 - 3\delta/2$; so overall Algorithm 1 is $(2\epsilon, 2\delta)$-DP. $\qquad\square$

## B Missing Proofs — Utility

### B.1 Proof of Utility - Rudimentary Claims

Recall that in Part II we partition the domain into bins by using a list of sorted samples, and see if the examples in those bins pass our $L_2$ test – TestCloseness. Moreover should TestCloseness return NULL we unify bins, using the same sets of points. Throughout the analysis we assume that we know the endpoint of each bin, namely that the result of the Bernoulli r.v.s have been disclosed. Now that each bin $\Pi_i^j$ has $t_i$ points we argue that the distribution of the number of points following a binomial distribution from $\mathcal{P}$ (resp. $\mathcal{Q}$).

**Claim 16.** *Fix index $j$. Fix index $i$. Denote $I$ as the interval $(x_{(\pi_i^j)}, x_{(\pi_{i+1}^j)})$ which is some interval defining a bin in the partition $\Pi_i^j$, which contains $t_i$ points overall. Then the number of points in an interval $\Pi_i^j$ coming from $\mathcal{P}$ follow $\mathrm{Bin}\left(t_i, \frac{\mathcal{P}(I)}{\mathcal{P}(I)+\mathcal{Q}(I)}\right)$. Moreover, for any two disjoint bins $\Pi_i^j$ and $\Pi_{i'}^j$ it holds that the number of points from $\mathcal{P}$ (resp. $\mathcal{Q}$) in each bin is independent of the other.*

*Proof.* We draw $s$ from a Poisson distribution with parameter $N$, representing the number of samples we take from $\frac{1}{2}(\mathcal{P} + \mathcal{Q})$. As a result, we can determine that within any fixed interval, including the interval $I$, the number of points we denote as $\tilde{X}_I$ follows a Poisson distribution with parameter $\frac{1}{2}(\mathcal{P}(I))$. However, we know that the number of points within interval $I$ is exactly $t_i$.

So now, conduct a thought experiment, where we re-sample $S$ until precisely $t_i$ points from $S$ fall in $I$. Replacing our original $t$ points in $\Pi_i^0$ with the new points is indistinguishable as they are distributed precisely the same. Denote $\tilde{X}_i$ (resp. $\tilde{Y}_i$) as the number of points from $\mathcal{P}$ (resp. $\mathcal{Q}$) in our repeated thought experiment that fall in $I$, and define $X_I$ (resp. $Y_I$) as the number of points from $\mathcal{P}$ (resp. $\mathcal{Q}$) that fall within interval $I$ in the draw where precisely $t_i$ points fall in $I$ from both distributions together. Therefore

$$X_I \sim \tilde{X}_I \mid \left( \tilde{X}_I + \tilde{Y}_I = t_i \right)$$

We can recall that by the definition of our sampling from Poisson distribution, we get that $\tilde{X}_I \sim \text{Poi}\left(\frac{N}{2}\mathcal{P}(I)\right)$, $\tilde{Y}_I \sim \text{Poi}\left(\frac{N}{2}\mathcal{Q}(I)\right)$, hence by Proposition 2 it follows that $X_I \sim \text{Bin}(t_i, \frac{\mathcal{P}(I)}{\mathcal{P}(I)+\mathcal{Q}(I)})$. Note also that $X_I$ and $X_{I'}$ are independent for every $I$ and $I'$. This is due to the fact that the number of points in any given interval, taken from the set $\mathcal{P}$, follows a Poisson distribution. As a result, the number of points in any two disjoint intervals $I$ and $I'$ are independent. $\square$

We now prove that our new selection of points in each bin follows a Poisson distribution.

**Claim 17.** *For each interval $I$ defining a bin $\Pi_i^j$, as in Claim 16, it holds that $X_i^j \sim \text{Poi}\left(\frac{N\mathcal{P}(I)}{4m(\mathcal{P}(I)+\mathcal{Q}(I))}\right)$ and $Y_i^j \sim \text{Poi}\left(\frac{N\mathcal{Q}(I)}{4m(\mathcal{P}(I)+\mathcal{Q}(I))}\right)$*

*Proof.* Using the same notation as in Claim 16 — we know that for each interval $I$ defining a bin $\Pi_i^j$, the following holds: $X_I$ is distributed according to a binomial distribution with parameters $t_i$ and $\frac{\mathcal{P}(I)}{\mathcal{P}(I)+\mathcal{Q}(I)}$. Now, let us take a subset of points $C_i^j$ of size $N_i^j$ from $I$, where $N_i^j$ is a Poisson random variable with parameter $\lambda = \frac{N}{4m^j}$, and denote $X_i^j$ as the sum of Bernoulli i.i.d. random variables $b_r$, such that each $b_r$ is distributed according to a Bernoulli distribution $\text{Ber}(\frac{\mathcal{P}(I)}{\mathcal{P}(I)+\mathcal{Q}(I)})$. In other words, $X_i^j = \sum_{r=1}^{N_i} b_r$. Based on Proposition 2 in the preliminaries, it holds that $X_i^j$ is distributed like a Poisson r.v. with parameter $\lambda \cdot \frac{\mathcal{P}(I)}{\mathcal{P}(I)+\mathcal{Q}(I)}$. The proof for $Y_i^j$ is symmetrical. To prove independence of $X_i^j = \sum_{i=1}^{N_i^j} \text{Ber}(p)$ and $Y_i^j = \sum_{i=1}^{N_i^j} \text{Ber}(1-p)$, and we have

$$
\begin{aligned}
\Pr[X_i = a, Y_i = b] &= \Pr[X_i = a, X_i + Y_i = a + b] \\
&= \Pr[X_i = a \mid X_i + Y_i = a + b] \cdot \Pr[X_i + Y_i = a + b] \\
&= \binom{a+b}{a} p^a (1-p)^b \cdot \frac{e^{-\left(\frac{N}{4m}\right)} \left(\frac{N}{4m}\right)^{a+b}}{(a+b)!} \\
&= \frac{1}{a!b!} p^a (1-p)^b \cdot \frac{e^{-\left(\frac{N}{4m}\right)p + \left(\frac{N}{4m}\right)(1-p)} \left(\frac{N}{4m}\right)^{a+b}}{(a+b)!} \\
&= \frac{\left(\frac{N}{4m}p\right)^a e^{-\left(\frac{N}{4m}\right)p}}{a!} \cdot \frac{\left(\frac{N}{4m}(1-p)\right)^b e^{-\left(\frac{N}{4m}\right)(1-p)}}{b!} = \Pr[X_i = a] \cdot \Pr[Y_i = b] \square
\end{aligned}
$$

Moreover, we need to ensure that each selection of a subset of points does not require more points than we already have, which is no more than $\frac{N}{m}$ for each bin, meaning no bin holds any $\perp$ points.

**Claim 18.** *If $N > \frac{32}{3} \cdot m \ln(2m)$, then with probability $1 - \frac{1}{2m}$ it holds that for each bin $i$, the size of the chosen subset $C_i$ is no greater than $\frac{N}{m} - 1$.*

*Proof.* The proof follows from standard tail bounds of the Poisson distribution. Given a random variable $X \sim \text{Poi}(\lambda)$, we have

$$\Pr[X > \lambda + 3\lambda] \leq \exp\left(-\frac{(3\lambda)^2}{3 \cdot (3+1)\lambda}\right) \leq \exp\left(-\frac{3\lambda}{4}\right)$$

Therefore, if we want each of the $m^0 + m^1 + m^2 + ... + m^j \leq 2m$ bins to have a subset size that doesn't exceed $\frac{N}{m^j}$, we get

$$\Pr\left[\forall i, \forall j, \ |C_i^j| < \frac{N}{m^j}\right] \geq 1 - \sum_{i,j} \Pr[N_i^j < 4\frac{N}{4m^j}] \geq 1 - \sum_j m^j \exp\left(-\frac{3N}{16m^j}\right)$$

$$\geq 1 - 2m \exp\left(-\frac{3N}{16m^0}\right) = 1 - 2me^{-2\ln(2m)} = 1 - \frac{1}{2m} \qquad \square$$

**Lemma 19** (Lemma 9 restated). *Fix index $j$. For each index $i$ denote by $I_i$ as the interval $(x_{(\pi_i^j)}, x_{(\pi_{i+1}^j)})$ which is the interval defining bin $i$ in the partition $\bar{\Pi}^j$. Then the estimator $Z$ computed by* TestCloseness *satisfies that $\mathbb{E}[Z] = \sum_{i=1}^{m^j}(p_i - q_i)^2$ where*

$$p_i = \frac{N\mathcal{P}(I_i)}{4m(\mathcal{P}(I_i) + \mathcal{Q}(I_i))}, \qquad q_i = \frac{N\mathcal{Q}(I_i)}{4m(\mathcal{P}(I_i) + \mathcal{Q}(I_i))}.$$

*and that $Var[Z] = \sum_{i=1}^{m^j} 4(p_i + q_i)(p_i - q_i)^2 + 2(p_i + q_i)^2$.*

*Proof.* For brevity, we denote $X_i^j$ as $X_i$ and the same for $Y_i$. We want to calculate the expectation of the statistic $Z = \sum_i (X_i - Y_i)^2 - X_i - Y_i$. Therefore

$$\mathbb{E}[Z] = \mathbb{E}\left[\sum_{i=1}^{m^j}(X_i - Y_i)^2 - X_i - Y_i\right] = \sum_{i=1}^{m^j} \mathbb{E}[X_i^2 - 2X_iY_i + Y_i^2 - X_i - Y_i]$$

$$= \sum_{i=1}^{m^j} p_i + p_i^2 - 2p_iq_i + q_i + q_i^2 - p_i - q_i = \sum_{i=1}^{m^j}(p_i - q_i)^2$$

We now want to calculate the variance of $Z = \sum_{i=1}^{n^j}(X_i - Y_i)^2 - X_i - Y_i$. Therefore

$$Var[Z] = Var\left[\sum_{i=1}^{n^j}(X_i - Y_i)^2 - X_i - Y_i\right] = \sum_{i=1}^{n^j} Var\left[(X_i - Y_i)^2 - X_i - Y_i\right]$$

Let's denote $Z_i = (X_i - Y_i)^2 - X_i - Y_i$ and calculate $\mathbb{E}\left[Z_i^2\right]$:

$$\mathbb{E}\left[Z_i^2\right] = \mathbb{E}\left[\left((X_i - Y_i)^2 - X_i - Y_i\right)^2\right] = \mathbb{E}\left[(X_i - Y_i)^4 - 2(X_i + Y_i)(X_i - Y_i)^2 + (X_i + Y_i)^2\right]$$

We analyze the expectations of each term separately, based on known first to fourth moments of the Poisson distribution [25] and the independence of $X_i$ and $Y_i$, we know that

$$\mathbb{E}\left[(X_i - Y_i)^4\right] = \mathbb{E}[X_i^4 - 4X_i^3Y_i + 6X_i^2Y_i^2 - 4X_iY_3 + Y_i^4]$$
$$= p_i^4 + 6p_i^3 + 7p_i^2 + p_i - 4p_i(q_i^3 + 3q_i^2 + q_i) + 6(p_i + p_i^2)(q_i + q_i^2)$$
$$- 4q_i(p_i^3 + 3p_i^2 + p_i) + q_i^4 + 6q_i^3 + 7q_i^2 + q_i$$
$$= (p_i - q_i)^4 + 6p_i^3 + 7p_i^2 + p_i - 4p_i(3q_i^2 + q_i) + 6p_iq_i + 6p_iq_i^2 + 6p_i^2q_i$$
$$- 4q_i(3p_i^2 + p_i) + 6q_i^3 + 7q_i^2 + q_i$$
$$= (p_i - q_i)^4 + 7(p_i - q_i)^2 + 6(p_i^3 - p_i^2q_i - p_iq_i^2 + q_i^3) + p_i + q_i + 12p_iq_i$$

It is to see that $\mathbb{E}[(X_i + Y_i)^2] = p_i + p_i^2 + q_i + q_i^2 + 2p_iq_i = (p_i + q_i)^2 + p_i + q_i$, so now

$$\mathbb{E}[(X_i + Y_i)(X_i - Y_i)^2] = \mathbb{E}[X_i^3 - 2X_i^2Y_i + X_iY_i^2 + Y_iX_i^2 - 2X_iY_i^2 + Y_i^3]$$
$$= p_i^3 + 3p_i^2 + p_i - q_i(p_i + p_i^2) - p_i(q_i + q_i^2) + q_i^3 + 3q_i^2 + q_i$$
$$= p_i^3 - q_ip_i^2 - p_iq_i^2 + q_i^3 + 3(p_i - q_i)^2 + p_i + q_i + 4p_iq_i$$

Combining all three terms, we get

$$\mathbb{E}\left[Z_i^2\right] = (p_i - q_i)^4 + 7(p_i - q_i)^2 + 6(p_i^3 - p_i^2 q_i - p_i q_i^2 + q_i^3) + p_i + q_i + 12 p_i q_i$$
$$- 2(p_i^3 - q_i p_i^2 - p_i q_i^2 + q_i^3 + 3(p_i - q_i)^2 + p_i - q_i + 4 p_i q_i) + (p_i + q_i)^2 + p_i + q_i$$
$$= (p_i - q_i)^4 + 4(p_i^3 - p_i^2 q_i - p_i q_i^2 + q_i^3) + (p_i - q_i)^2 + (p_i + q_i)^2 + 4 p_i q_i$$
$$= (p_i - q_i)^4 + 4(p_i + q_i)(p_i - q_i)^2 + 2(p_i + q_i)^2$$

And so we get that

$$\mathrm{Var}[Z_i] = \mathbb{E}[Z_i^2] - \mathbb{E}[Z_i]^2 = (p_i - q_i)^4 + 4(p_i + q_i)(p_i - q_i)^2 + 2(p_i + q_i)^2 - (p_i + q_i)^4$$
$$= 4(p_i + q_i)(p_i - q_i)^2 + 2(p_i + q_i)^2$$

So overall by independence we get

$$\mathrm{Var}[Z] = \sum_{i=1}^{m^j} \mathrm{Var}[Z_i] = \sum_{i=1}^{m^j} 4(p_i + q_i)(p_i - q_i)^2 + 2(p_i + q_i)^2 \qquad \square$$

**Claim 20** (Claim 10 restated). *If $N > 3000 m \log(2m)$, then with a probability of $1 - \frac{1}{m}$, any $I$ that forms a bin $\Pi_i^0$ in $\overline{\Pi}^0$, has that $\frac{1}{1.01m} \leq \frac{\mathcal{P}(I) + \mathcal{Q}(I)}{2} \leq \frac{1.01}{m}$.*

*Proof.* Let $I$ be an interval determining a bin $\Pi_i^0$, and denote $S_I = \frac{1}{2}(\mathcal{P}(I) + \mathcal{Q}(I))$. From our process of partitioning the domain and creating the intervals, we know that $S_I \sim \sum_{i=1}^{N/m} \mathrm{Exp}(N)$. Therefore, as the sum of independent Exponential r.v.s we can conclude that $S_I \sim \mathrm{Gamma}\left(N/m, \frac{1}{N}\right)$. Using the tail bound of the sum of the exponential distribution [24], we get the following inequality:

$$\forall t > 0, \ \Pr[S_I > t\mathbb{E}[S_I]] \leq \min\left\{\frac{1}{t}, 1\right\} \exp(-\alpha(t - 1 - \log(t)))$$

where $\alpha = \frac{\mathbb{E}[S_I]}{1/N}$. In our case, we have $\mathbb{E}[S_I] = \frac{N/m}{N}$, so $\alpha = \frac{N}{m}$. By setting $t = 1.01$, we arrive at the following result:

$$\Pr\left[S_I > \frac{1.01}{m}\right] = \Pr\left[S_I > \frac{1.01N}{m\alpha} \cdot \frac{1}{m}\right] \leq \frac{1}{1.01} e^{-\frac{N}{m}(1.01 - 1 - \log(1.01))} < e^{-\frac{0.00049N}{m}} \leq \frac{1}{2m^2}$$

And now if $t = \frac{1}{1.01} \leq 1$, then we get

$$\Pr\left[S_I \leq \frac{1}{1.01m}\right] \leq e^{-\frac{N}{m}(1/1.01 - 1 - \log(1/1.01))} \leq e^{-\frac{0.00049N}{m}} \leq \frac{1}{2m^2}$$

So by the Union-Bound, we get that $\Pr[\exists I, \ S_I > 1.01/m \text{ or } S_I < 1/1.01m] \leq m\left(\frac{1}{2m^2} + \frac{1}{2m^2}\right) = \frac{1}{m}$.
$\square$

### B.1.1  Proof of Theorem 7

Having acquired the rudimentary tools, we can now prove Theorem 7.

*Proof of Theorem 7.* We can see that in the Part I of the algorithm, we have $s \sim \mathrm{Poi}(m)$ samples such that for all $1 \leq i \leq s$ $\ell(x_i) \in \{1, -1\}$ with equal probability and independent from each other, therefore, by linearity of expectation we get that for the estimator $Z$ from Line 5 of Part I of Algorithm 1 we have $\mathbb{E}[Z] = \sum_i \mathbb{E}[\ell(x_i)\ell(x_{i+1})] = \sum_i \mathbb{E}[\ell(x_i)]\mathbb{E}[\ell(x_{i+1})] = 0$. We can bound the variance of $Z$ using $O(m)$ by the Effron Stein inequality / Jackknife principle.

**Proposition 21.** *$\mathrm{Var}[Z] \leq 9m$.*

*Proof of Proposition 21.* Denote our points $X = (x_1, x_2, \ldots, x_i, \ldots, x_s) \in \{1, -1\}^s$, and it is clear to see that if we change one of the points independently $X^{(i)} = (x_1, x_2, \ldots, x_i', \ldots, x_s)$ then

$$\mathrm{Var}[Z|s] \leq \frac{1}{2} \sum_{i=1}^s \mathbb{E}\left[\left(Z(X) - Z\left(X^{(i)}\right)\right)^2\right]$$

$$\leq \frac{1}{2} \sum_{i=1}^s \mathbb{E}\left[(\ell(x_{i-1})\ell(x_i) + \ell(x_i)\ell(x_{i+1}) - \ell(x_{i-1})\ell(x_i') - \ell(x_i')\ell(x_{i+1}))^2\right] \leq \frac{1}{2} \cdot 16s = 8s$$

And we know that $\text{Var}[\mathbb{E}[Z|s]] \leq \text{Var}[s] = m$. Therefore we get

$$\text{Var}[Z] = \mathbb{E}[\text{Var}[Z|s]] + \text{Var}[\mathbb{E}[Z|s]] \leq \mathbb{E}[8s] + m = 9m \qquad \square$$

Therefore by Chebyshev's inequality, it follows that

$$\Pr[Z \leq 10\sqrt{m}] \geq 1 - \Pr[|Z - \mathbb{E}[Z]| > 10\sqrt{m}] \geq 1 - \frac{9m}{100m} \geq 0.91.$$

Now we prove that our Laplace noise does not exceed $\frac{m^3\alpha^3}{8k^2}$, provided $m \geq \frac{100k^{\frac{2}{3}}}{\alpha\epsilon^{\frac{1}{3}}}$.

$$\Pr[|\tilde{Z} - Z| \geq \frac{1}{4}\frac{m^3\alpha^3}{2k^2}] = \Pr\left[\text{Lap}\left(\frac{6}{\epsilon}\right) \geq \frac{1}{4}\frac{m^3\alpha^3}{2k^2}\right] \leq \exp\left(-\frac{m^3\alpha^3\epsilon}{48k^2}\right) \leq 0.01 \qquad (1)$$

And so , with probability $\geq 0.9$ it holds that $\tilde{Z} \leq 10\sqrt{m} + \frac{1}{4}\frac{m^3\alpha^3}{2k^2} < \frac{m^3\alpha^3}{2k^2}$, provided $m \geq 40\frac{k^{\frac{4}{5}}}{\alpha^{\frac{6}{5}}} + 100\frac{k^{\frac{2}{3}}}{\alpha\epsilon^{\frac{1}{3}}}$. It follows that Part I of Algorithm 1 returns NULL.

In Part II, we also need to argue that all invocations of TestClosenessreturn NULL. Fix an iteration $j$ and observe that for any $i$ it holds that $X_i^j$ and $Y_i^j$ are distributed like $\text{Poi}(\frac{N}{8m^j})$ as the labeling of points as coming from $\mathcal{P}$ or $\mathcal{Q}$ is completely random $\text{Ber}(1/2)$. We next bound $|X_i^j - Y_i^j|$ and $\sum_i (X_i^j - Y_i^j))^2$ using known tail bounds on Poisson random variables (Proposition2).

**Proposition 22.** *Assume that for all $j$ we have $N/8m^j > 4\log(800m)$. Then with probability $> 0.99$ it holds that for any $j$ and any $i$ we have*

$$|X_i^j - Y_i^j| \leq \sqrt{\frac{6N}{m^j}\log(800m)}$$

*Proof.* We have that $N_i^j \sim \text{Poi}(\frac{N}{4m^j})$, $\text{Poi}(\frac{N}{8m^j})$ and $Y_i = N_i^j - X_i^j$. Therefore, $|X_i^j - Y_i^j| = |2X_i^j - N_i^j| \leq 2|X_i^j - \mathbb{E}[X_i^j]| + |N_i^j - \mathbb{E}[N_i^j]| + |2\mathbb{E}[X_i^j] - \mathbb{E}[N_i^j]|$. Seeing as $2\mathbb{E}[X_i^j] - \mathbb{E}[N_i^j] = 0$ we bound the other two terms using known tail bounds on the Poisson distribution from Proposition 2.

$$\Pr[|X_i^j - \mathbb{E}[X_i^j]| > 2\sqrt{N/8m^j \log(800m)}] < \frac{1}{400m}$$
$$\Pr[|N_i^j - \mathbb{E}[N_i^j]| > 2\sqrt{N/4m^j \log(800m)}] < \frac{1}{400m}$$

Provided $N/8m^j > 4\log(800m)$. And so, w.p. $\geq 1 - \frac{1}{200m}$ it follows that $|X_i^j - Y_i^j| \leq 4\sqrt{N/8m^j \log(800m)} + 2\sqrt{N/4m^j \log(800m)} \leq \sqrt{6N/m^j \log(800m)}$. Applying the Union Bound on all $m^0 + m^1 + ... + m^{j_0} \leq 2m$ bins, we have the required. $\square$

We now can complete the proof that Part II also returns NULL. Note that in each invocation of TestClosenesswe use $m = \sum_j N_i^j$ points over $n = m^j$ bins. It was already established in the proof of Lemma 6 that the probability that there exists an invocation where $n_{\max} \leq \sqrt{\frac{6m}{n}\log(800m)}$ yet due to the Laplace noise $\hat{n}_{\max} > \sqrt{\frac{6m}{n}\log(800m)} + \frac{8\log(2/j_0\delta)}{\epsilon'}$ is upper bounded by $\frac{\delta}{2}$. We show a similar result for the difference of $\hat{Z} - Z$, which we denote as a random variable $R \sim \text{Lap}(16\log_{4/3}(2/\delta)n_z/\epsilon')$. Standard properties of the Laplace distribution give that the probability that exists even a single invocation of TestCloseness where $R$ is greater than $\log(10j_0)16\log_{4/3}(2/\delta)n_z/\epsilon'$ is at most 0.01. Lastly, we argue that $Z$ isn't too large. Based on Lemma 9 we know that at any iteration $j$ we have $\mathbb{E}[Z] = \sum_{i=1}^{m^j}(\frac{N}{8m} - \frac{N}{8m}) = 0$ and $\text{Var}[Z] \leq 2 \cdot (N/8m^j + N/8m^j)^2 m^j = \frac{N^2}{8m^j}$. Using the Chebyshev Inequality and the Union Bound that $\Pr[\exists j, \ Z > 10N\sqrt{\log(1/\alpha)/8m^j}] \leq j_0 \cdot \frac{1}{100\log(1/\alpha)} < 0.01$.

And so, w.p. $\geq 0.97 - \delta \geq 0.96$ we get that

$$\hat{Z} = Z + R \leq 10N\sqrt{\log(1/\alpha)/8m^j} + \frac{16\log(10j_0)\log_{4/3}(2/\delta)n_z}{\epsilon'}$$

$$\overset{(*)}{\leq} 10N\sqrt{\log(1/\alpha)/8m^j} + \frac{(\frac{\alpha}{12\sqrt{2k+1}\cdot\log(1/\alpha)})^2 N^2}{16}$$

$$\overset{(**)}{\leq} \frac{(\frac{\alpha}{12\sqrt{2k+1}\cdot\log(1/\alpha)})^2 N^2}{16} + \frac{\alpha^2 N^2}{16} = \frac{(\frac{\alpha}{12\sqrt{k+1}\cdot\log(1/\alpha)})^2 N^2}{8} \overset{(***)}{\leq} \frac{1}{2}\alpha^2_{\mathsf{TestCloseness}}(\sum_i N_i^j)$$

Where inequality $(*)$ holds when

$$\frac{16\log(10j_0)\log_{4/3}(2/\delta)\left(\sqrt{\frac{6m}{n}\log(800m)} + \frac{256\sqrt{j_0\log(2/\delta)}\ln(2j_0/\delta)}{\epsilon}\right)}{\frac{\epsilon}{8\sqrt{j_0\log(2/\delta)}}} < \frac{(\frac{\alpha}{12\sqrt{2k+1}\cdot\log(1/\alpha)})^2 N^2}{16} \Leftrightarrow$$

$$\frac{144\sqrt{\log(1/\alpha)\log(2/\delta)}\log(10\log(1/\alpha))\log_{4/3}(2/\delta)\left(\sqrt{\frac{12N}{m^j}\log(800m)} + \frac{256\sqrt{\log(1/\alpha)\log(2/\delta)}\ln(2j_0/\delta)}{\epsilon}\right)}{\epsilon} < \frac{\alpha^2 N^2}{6912k\log^2(1/\alpha)}$$

which in turn holds when $N \geq \tilde{\Omega}(\frac{k^{1/3}}{\epsilon^{2/3}\alpha^{4/3}})$ and $N \geq \tilde{\Omega}(\frac{k^{1/2}}{\epsilon\alpha})$;
inequality $(**)$ holds when

$$10N\sqrt{\frac{\log(1/\alpha)}{8m^j}} \leq 10N\sqrt{\frac{\log(1/\alpha)}{8k}} \leq \frac{\alpha^2 N^2}{6912k\log^2(1/\alpha)}$$

which in turn holds when $N \geq \Omega(\frac{k^{1/2}\log(1/\alpha)^{5/2}}{\alpha^2})$; and inequality $(***)$ holds due to Proposition 22
when

$$\sum_i N_i^j \geq N - m^j \cdot 2\sqrt{N/4m^j\log(800m)} \geq N/2$$

when $N \geq 3000m\log(800m)$. $\qquad\qquad\square$

## B.2  Proof of Theorem 8

**Lemma 23** (Lemma 11 restated). *Fix $p \in (0,1)$. Suppose there exists $n$ non-negative random variables $X_1, X_2, \ldots, X_n$, such that for each $i$ it holds that for some fixed number $a_i$ we have $\Pr[X_i \geq a_i] \geq p$. Then, given a constant $c > 0$ there exists another constant $C > 0$ such that with a probability at least $C$ it holds that $\sum_{i=1}^n X_i \geq c\sum_{i=1}^n a_i$.*

*Proof.* Define the random variables $Y_i = \begin{cases} a_i & \text{with probability } p \\ 0 & \text{with probability } 1-p \end{cases}$. We can see that
$\mathbb{E}\left[\sum_{i=1}^n Y_i\right] = p\sum_{i=1}^n a_i$ and that the variance is $\text{Var}\left[\sum_{i=1}^n Y_i\right] = \sum_{i=1}^n \text{Var}[Y_i] = p(1-p)\sum_{i=1}^n a_i^2$. Fix $c > 0$. Using Chebyshev's inequality, we can bound the probability of being far from their expectation.

$$\Pr\left[\sum_{i=1}^n Y_i \leq c \cdot \mathbb{E}\left[\sum_{i=1}^n Y_i\right]\right] \leq \Pr\left[\left|\sum_{i=1}^n Y_i - \mathbb{E}\left[\sum_{i=1}^n Y_i\right]\right| \geq (1-c)\mathbb{E}\left[\sum_{i=1}^n Y_i\right]\right]$$

$$\leq \frac{\text{Var}\left[\sum_{i=1}^n Y_i\right]}{(1-c)^2 \cdot \mathbb{E}\left[\sum_{i=1}^n Y_i\right]^2} \leq \frac{p(1-p)\sum_{i=1}^n a_i^2}{(1-c)^2 p^2 \left(\sum_{i=1}^n a_i^2\right)^2} \leq \frac{p(1-p)}{p^2(1-c)^2}$$

Now, since we have $\Pr[X_i \geq a_i] \geq p = \Pr[Y_i = a_i]$ and $\Pr[X_i \in (0, a_i)] \leq 1 - p = \Pr[Y_i = 0]$ for each $i$, it is easy to see that $\Pr[\sum_i^n X_i \geq a] \geq \Pr[\sum_i^n Y_i \geq a]$ for any $a$. Hence

$$\Pr\left[\sum_{i=1}^n X_i \geq c\sum_{i=1}^n a_i\right] \geq \Pr\left[\sum_i^n Y_i \geq c\sum_{i=1}^n a_i\right] \geq 1 - \frac{p(1-p)}{p^2(1-c)^2} = C \qquad\square$$

**Lemma 24.** *[17] For any two distributions $\mathcal{P}$ and $\mathcal{Q}$ on $[m]$, let $\mathcal{P}'$ and $\mathcal{Q}'$ be the merged distributions, Then,*

$$\|\mathcal{P} - \mathcal{Q}\|_{\mathcal{A}_k} \leq \|\mathcal{P}' - \mathcal{Q}'\|_{\mathcal{A}_k} + 2\|\mathcal{P} - \mathcal{Q}\|_{1,k}$$

*Proof.* Let $\mathcal{I}$ be the partition of $[m]$ into $k$ intervals so that $\|\mathcal{P} - \mathcal{Q}\|_{\mathcal{A}_k} = \sum_{i \in \mathcal{I}} |\mathcal{P}(I) - \mathcal{Q}(I)|$. Let $\mathcal{I}'$ be obtained from $\mathcal{I}$ by rounding each upper endpoint of each interval (except for the last) down to the nearest even integer, and rounding the lower endpoint of each interval up to the nearest odd integer. Note that

$$\sum_{I \in \mathcal{I}'} |\mathcal{P}(I) - \mathcal{Q}(I)| = \sum_{I \in \mathcal{I}'} |\mathcal{P}'(I/2) - \mathcal{Q}'(I/2)| \leq \|\mathcal{P}' - \mathcal{Q}'\|_{\mathcal{A}_k}$$

seeing as the partition $\mathcal{I}'$ is obtained from $\mathcal{I}$ by taking at most $k$ points and moving them from one interval to another. Therefore, the difference $\left| \sum_{I \in \mathcal{I}} |\mathcal{P}(I) - \mathcal{Q}(I)| - \sum_{I \in \mathcal{I}'} |\mathcal{P}(I) - \mathcal{Q}(I)| \right|$ is at most twice the sum of $|\mathcal{P}(i) - \mathcal{Q}(i)|$ over these $k$ points, and therefore at most $2\|\mathcal{P} - \mathcal{Q}\|_{1,k}$. Combining this with the above gives our result. $\qquad\square$

**Lemma 25** (Lemma 13 restated). *[17] For any two distributions $\mathcal{P}$ and $\mathcal{Q}$ on $[m]$ such that $\|\mathcal{P} - \mathcal{Q}\|_{\mathcal{A}_k} > \alpha$, there iteration $j \in [\log(m/k)]$ such that $\|\mathcal{P}^{\Pi^j} - \mathcal{Q}^{\Pi^j}\|_{1,k} > \alpha/\log(m/k)$.*

*Proof.* Lemma 24 asserts that

$$\alpha < \|\mathcal{P} - \mathcal{Q}\|_{\mathcal{A}_k} \leq \|\mathcal{P}^{\Pi^1} - \mathcal{Q}^{\Pi^1}\|_{\mathcal{A}_k} + 2\|\mathcal{P}^{\Pi^0} - \mathcal{Q}^{\Pi^0}\|_{1,k}$$

We apply this recursively when we know that in the final level $j_0 = \log(m/k)$, we get that $\|\mathcal{P}^{\Pi^j} - \mathcal{Q}^{\Pi^j}\|_{\mathcal{A}_k} = \|\mathcal{P}^{\Pi^j} - \mathcal{Q}^{\Pi^j}\|_{1,k}$ because the distribution there has at most $k$ bins. Thus

$$\sum_{j=1}^{\log(m/k)} \|\mathcal{P}^{\Pi^j} - \mathcal{Q}^{\Pi^j}\|_{1,k} \geq \alpha$$

Therefore, by the average principle one of the $j \in [\log(m/k)]$ must satisfy $\|\mathcal{P}^{\Pi^j} - \mathcal{Q}^{\Pi^j}\|_{1,k} \geq \alpha/\log(m/k)$; which, by Cauchy–Schwarz, gives $\|\mathcal{P}^{\Pi^j} - \mathcal{Q}^{\Pi^j}\|_2 \geq \alpha / \left(\sqrt{k}\log(m/k)\right)$. $\qquad\square$

