# OpenReview forum: "Differentially Private Equivalence Testing for Continuous Distributions and Applications"
_NeurIPS.cc/2024/Conference — NeurIPS 2024 poster_

### Official Review · Reviewer_3bTx · 2024-06-29

**Soundness:** 3
**Presentation:** 2
**Contribution:** 2
**Rating:** 7
**Confidence:** 3

**Summary:**

This paper studies closeness (equivalent) testing between two continous distributions under approximate differential privacy. In particular, they propose a private version of the equivalent testing algorithm in [Diakonikolas et al.], which discerns whether the two are identical or far-apart in terms of the $\mathcal{A}_k$ distance. The structure of the DP algorithm in this paper basically follows that of the algorithm in [Diakonikolas et al.]. The first stage is a simple testing algorithm that counts the difference between pairs of successive samples coming from the same distribution and pairs of successive samples from different distributions. In the second stage, the algorithm divides data points into $m$ intervals to take sufficient advantage of these intervals. In particular, it repeatedly runs a closeness tester based on $\ell_2$ norm, and merge the bins after each iterations.

To privatize the above algorithm, there are two main technical obstacles. First is that the private algorithm no longer be able to re-sample new points to estimate the $\ell_2$ distances. To adress this, the algorithm in this paper uses Poisson-drawn subsamples of each bin and revisit the utility analysis of [Diakonikolas et al.]. The second obstacle is that changing a datapoint might shift all bins in the worst case. To bypass this issue, this paper uses a simple coupling argument. In particular, their algorithm "randomizes" the size of each bin by an independent Bernoulli random variable to "correct" such shifting.

This paper also gives the applications of their algorithm on multiple families of distributions. Their approach can be easily applied to discrete distributions.

**Strengths:**

This paper continues the line of work on designing DP-hypothesis testers, and give the first algorithm on privately testing the equivalence for continous distributions. Although the construction of tester in this paper basically follows the structure in [Diakonikolas et al.], but I appreciate that the authors repeatdly and accurately elaborate on which part of their proof technique deviates from that in [Diakonikolas et al.]. In particular, the coupling argument on the Bernoulli random variables to bound the sensitivity of shifts is simple but sweet. All theorems are clearly stated and the proofs are correct.

**Weaknesses:**

I have a bit concern about the presentation of this paper. For example, the authors do not introduce their privacy notion (that is, the definition of "neighboring datasets") at all. I believe it is important to clarify this because, at first glance, it appears that the input to the algorithm is distributions rather than datasets, and I feel frustated that I have to guess what is the definition of "neighboring" by reading the proof the Theorem 3.

Some typos:
1. In line 10 and line 11 of Algorithm 1, should the letter 'j' be lowercase?
2. In line 188, a comma was mistakenly written as a period.

**Questions:**

Compared to designing DP-equivalence testers for discrete distributions, what is the most fundamental technical challenge in doing so for continuous distributions? Is this challenge inherited from the corresponding non-private algorithms?

**Limitations:**

This paper discusses several limitations.

---

> ### Author Rebuttal · Authors · 2024-08-06
>
> 1. The main challenge in transitioning a non-private algorithm to a private one in a continuous setting is using the \emph{data itself} to divide the domain. As far as we know, there is no known algorithm for the continuous case in distribution testing that does not in some way partition the domain.  It is well-known that the problem of finding an interior point (outputting a point from a distribution within an interval) in general, without any assumptions, is impossible in the continuous case. Our solution to this is to alter the index of partitioning using Bernoulli rvs.

---

> > ### Comment · Reviewer_3bTx · 2024-08-11
> >
> > Thank you for the rebuttal.

---

### Official Review · Reviewer_ZYiV · 2024-07-13

**Soundness:** 3
**Presentation:** 2
**Contribution:** 2
**Rating:** 6
**Confidence:** 4

**Summary:**

This paper introduces a novel algorithm for equivalence testing between two continuous distributions under the framework of differential privacy. The proposed method adapts the algorithm by Diakonikolas et al. to a differentially private version, using various clever constructions and privacy mechanisms. The authors derive a theoretical guarantee for the algorithm in terms of a lower bound rate on the number of samples needed to correctly discern the null hypothesis that two distributions are equal versus the alternative that they are $\alpha$ apart in $\mathcal{A}_k$-norm. The bound is under the model characteristics $\alpha$, $k$ and the privacy parameter $\epsilon$.

**Strengths:**

* The authors present a novel and creative algorithm tackling an interesting general hypothesis testing problem.
* The exposition of the algorithm is clear and its logic is well explained, which is a feat considering its complexity.
* The authors provide a theoretical guarantee with sound proofs that are well-explained.
* The method provided by the authors should be rate optimal in most high dimensional, large sample applications, in which case the state of the art nonprivate rate of $\sqrt{k}/\alpha^2$.
* I enjoyed reading the article.

**Weaknesses:**

* The article lacks a broader discussion of the rate attained by the method. The rate attained by the algorithm consists of the maximum of 4 terms, where $\alpha$ should be considered small (e.g. decreasing as the sample size increases), and $k$ perhaps (very) large. The authors consider a large sample regime. In the current formulation, privacy in most cases, comes at no cost (e.g. the maximum is typically just taken in $\sqrt{k}/\alpha^2$)? Furthermore, do the phase transitions have any meaning specifically? Can anything be said about optimality here? To truly make this a strong contribution, I think an optimality result of some sorts is desirable.

* Related to the earlier point: How do the rates compare to those established for continuous distributions belonging to parametric families under DP? In particular, I would like to see mention/discussion of the works of for example: Private Identity Testing for High-Dimensional Distributions -- Canonne et al. (2020) or Private Identity Testing for High-Dimensional Distributions -- S. Narayanan (2022). Although the aforementioned papers consider a different setting, they have accompanying optimality results and provide grounds for assessment of the method of this paper as well. The authors also mention that the method applies to discrete distributions as well. How does the rate compare to those derived in discrete settings such as [2]?

Minor points:

* The way sampling is considered is somewhat ambiguously outlined, whilst this is key in any setting considering privacy. This seems to be a consequence of how the algorithm is designed / how its guarantees are shown. In a privacy setting, most naturally in my opinion, samples are fixed and given. Of course the formulated setting extends naturally to such a formulation for large enough samples, but I personally find the current formulation unnatural considering the privacy angle.

* Imprecision in keeping track of constants in the definition of the algorithm. Many constants are giving (ambiguously large) values (i.e. 10^7 in the algorithm itself). If one were to implement this algorithm, what values should one choose? Which ones are necessarily large in practice, and which are artifacts of the proof?

* The paper could use another thorough proofread for spelling and punctuation.

**Questions:**

Remarks / suggestions / questions:

* The notation $n$ is unexplained in the "Related Work" section. This should be the cardinality of the sample space when discussing [26] and others. Maybe $n$ is also a poor choice here, as this is typically used to denote sample size, where here it is more similar to dimensionality, or the role of $k$ in this article.

* When presenting the rate in the introduction (i.e. Fact 1) there seems to be some typos: $k^{4/5}/\alpha^{6/5}$ instead of $k^{4/5}/\alpha^{5/6} $ following the proof of Section 3. Same seems to be the case for the factor $k^{1/3}/(\alpha^{4/3} \epsilon^{2/3})$ in Fact 1; where the sample complexity of Algorithm 1 is of the order $k^{2/3}/(\alpha^{4/3} \epsilon^{1/3})$ following Section 3?

* The authors mention in the introduction that one could in principle run the private tester of Diakonikolas $O(1/\epsilon)$-times to attain the rate $\sqrt{k}/(\epsilon \alpha^2)$. This is however, only a lower bound, and for that quite a loose one. I do not see why noisy count would necessarily give this rate; it seems to be highly dependent on the power calculation used for each of the individual tests. Could this not be $\sqrt{k}/(\sqrt{\epsilon} \alpha^2)$, for example, considering how Binomial concentration?

**Limitations:**

What is missing limitations wise is a discussion of optimality, or of implementation of the algorithm in practice (e.g. computational complexity, practical range for $\epsilon$ for which the privacy constraints are impactful, choices of constants). Otherwise, the article does not have glaring limitations unless one goes beyond its current scope.

---

> ### Author Rebuttal · Authors · 2024-08-06
>
> Weakness question \#1:
> The constant $c_{dkn}$, which represents their list of inequality of the expectation with $\Omega$ notation in Diakonikolas et al, does not specify the exact value of this constant. In our second algorithm, we used $10^7$ because in their proof, Diakonikolas et al used the value $10^6$ for the constant for the analysis.
>
> Question \#1: In most of the papers referenced in the related work, $n$ is commonly used to denote the size of the domain. However, in our case, we do not have a specific size for the domain in the continuous regime. Instead, we use $k$ to represent the domain size that we intend to partition in the continuous distribution.
>
> Question \#2: Indeed, throughout the paper the true coefficient should be $\alpha^{-6/5}$. Similarly, the second term is indeed $k^{1/3}\alpha^{-4/3}\epsilon^{-2/3}$.
>
> Question \#3:
> When using the Subsample-and-Aggregate framework, running the non-private algorithm $O(1/\epsilon)$ times is an na\"ive upper bound which we haven't delved deeply into. Afterall, it is a baseline --- how'd I approach the problem had I not known how to privatize the algorithm of Diakonikolas et al. It might be the case that a tighter analysis of S\&A exists for Hypothesis Testing, but we are unaware of it.

---

> > ### Comment · Reviewer_ZYiV · 2024-08-11
> >
> > Thank you for your response. I wish to maintain my score.

---

### Official Review · Reviewer_XXUu · 2024-07-15

**Soundness:** 3
**Presentation:** 2
**Contribution:** 3
**Rating:** 6
**Confidence:** 3

**Summary:**

This paper considers the sample complexity of the problem of equivalence testing for continuous distributions under approximate differential privacy. Mathematically, given two distributions $P$ and $Q$ how many samples is required to have an algorithm that outputs $\texttt{yes}$ or $\texttt{no}$ such that

- if $P$ and $Q$ are equal, the algorithm outputs $\texttt{yes}$  with probability at least $2/3$, and
- if $P$ and $Q$ have distance at least $\alpha$ in $\mathcal{A}_k$-norm, the algorithm outputs $\texttt{no}$ with probability at least $2/3$.

Moreover, the algorithm must satisfy $(\varepsilon, \delta)$ differential privacy.

${\mathcal{A}} _ k$ norm restricts TV distance to using $k$-intervals:  $\lVert P - Q \rVert_{\mathcal{A} _ k} =  \sup{\mathcal{I}} \sum_{j=1}^k | P[I_j] - Q[I_j]|$, where $\mathcal{I}$ is a partition of $\mathbb{R}$ into $k$ intervals.

The sample complexity this paper obtains is

$$
\tilde{O} \left( \max \left\\{ \frac{k^{4/5}}{\alpha^{6/5}}, \frac{k^{1/2}}{\alpha^2}, \frac{k^{1/3}}{\alpha^{4/3} \varepsilon^{2/3}}, \frac{k^{1/2}}{\alpha \varepsilon} \right\\} \right).
$$

The first two terms are the non-private cost that matches the upper and lower bound of [1].

Technically this paper build upon the techniques of [1]. [1] considers the problem in the non private setting. This paper presents a privatization of the algorithm in [1], through a modification of the second phase of the algorithm in [1] to ensure low sensitivity.

[1] Ilias Diakonikolas, Daniel M Kane, and Vladimir Nikishkin. Optimal algorithms and lower bounds for testing closeness of structured distributions. In 2015 IEEE 56th Annual Symposium on Foundations of Computer Science, pages 1183–1202. IEEE, 2015.

**Strengths:**

Identity testing is a fundamental and conceptually important problem, and this paper presents the first algorithm for it in the continuous setting.

The paper modifies the algorithm from [1] to ensure low sensitivity, though further changes and analysis are required to ensure that these modifications do not cause issues for the analysis in [1].

**Weaknesses:**

The quality of writing and clarity could be improved at some parts. See questions, for some suggested changes.

**Questions:**

For the non-private part, [1] provides matching upper and lower bounds. Are lower bounds under privacy known for this problem? If not, what should we expect the correct bounds to be? It is mentioned that Acharya et al. provide lower bounds in the discrete setting. A comparison with those lower bounds might be helpful to demonstrate this.

In the main result, logarithmic factors are omitted, but I believe that in the privacy literature, $\log(1/\delta)$ is typically considered a polynomial factor. What is the dependence on $\delta$ in the sample complexity provided in this paper?

My understanding is that the binning of samples aids in the sensitivity analysis, while Poisson sampling facilitates the analysis from [1]. However, it is not clear which parts of the analysis are based on proofs from [1] and which parts are novel contributions.

Writing and clarity suggestions:
I think the description of [1]'s algorithm in 'our algorithm' section could be more detailed. I found understanding this paragraph a bit difficult, and I expect that a reader who is not familiar with [1] would benefit from a more comprehensive description of the algorithm. Since the rest of the algorithm relies on this part, and this is the only place it is explained in the text, it is crucial to provide a clear explanation, especially for lines 57-61.

I think there might be a typo on line 104. Another typo on line 161.

**Limitations:**

Yes, the authors have addressed the limitations.

---

> ### Author Rebuttal · Authors · 2024-08-06
>
> 1. We made no effort to minimize the polylog$(1/\delta)$ factor. The sample complexity is given in Line 1 of Part II of our algorithm: $N \gets 10^7\left(\frac{k^{1/3}}{\alpha^{4/3}\epsilon^{2/3}} + \frac {\sqrt k}{\alpha\epsilon}+\frac{\sqrt{k}}{\alpha^2}\right)\log^6(\frac k {\alpha\epsilon\delta})$.
>
> 2. ``My understanding is that the binning of samples aids in the sensitivity analysis, while Poisson sampling facilitates the analysis from [1].'' That understanding is spot-on.

---

> > ### Comment · Reviewer_XXUu · 2024-08-12
> >
> > I thank the authors for the rebuttal. I have updated my score.

---

### Official Review · Reviewer_aFH2 · 2024-07-15

**Soundness:** 3
**Presentation:** 2
**Contribution:** 3
**Rating:** 6
**Confidence:** 4

**Summary:**

This paper talks about differentially private mechanism for property testing -- testing if two continuous distributions are equivalent. The main contribution of this paper is to develop DP versions of the algorithm in [16], which does not support DP. The algorithms in [16] uses discretization and when two distributions are sufficiently different the discretized version has large L2 norms.

To support DP, there are challenges of developing the bucketing scheme. Instead of using a fixed bin size, the authors use sorted indices to define bins. This reduces data sensitivity. This requires new analysis of utility and privacy.

**Strengths:**

First the problem of DP equivalence testing is an interesting one. The algorithm is performing DP for the algorithm in [16]. Thus it is building on top of [16]. The algorithm makes sense and has merit.

**Weaknesses:**

There are a few issues that can be addressed to improve the paper.

It will be nice if the authors can present the algorithm 1 using plain English, instead of just presenting the pseudo code.

One thing that this paper can do better is to improve the discussion and comparison with the prior literature. In section 1.1 there are prior work on DP methods for identity testing and closeness testing. How does the algorithm in this paper compare with those?

Together with the issues above, it will be good to highlight the significance of contribution.

Line 45, what is alpha?

Line 58, the data using into -- drop either into or using

Line 72, use re-sample -- remove one of them

**Questions:**

See above

---

> ### Author Rebuttal · Authors · 2024-08-06
>
> 1. We tried to describe our algorithm prior to presenting it formally in lines 50-84. We would highly appreciate suggestions as to improving said description.
>
> 2. See above discussion as to lower bound comparison.
>
> 3. Alpha is the distance parameter. In our case is used for bound below the $\mathcal{A}_k$ distance under $\mathcal{H}_1$. (Theorem 8, line 209)

---

### Author Rebuttal · Authors · 2024-08-06

First we wish to thank all reviewers for their thoughtful remarks and some spot-on comments.

In broad brushstrokes, all reviews agree the paper and the algorithm has merit, but the presentation is lacking. We ourselves agree with the reviewer's feedback. In our defense we can only say that (1) the current version is far better then our initial draft; (2) we promise to use the additional page of the camera-ready version to implement the reviewers' suggestions and substantially improve the paper's presentation. In fact, some of the reviewers' comments are the direct result of brevity: we shrunk the Related Work section by removing lower bounds comparison and mentioning Identity Testing, and omitted the definition of Equivalence Testing and of neighboring instances from the Preliminaries hoping the reader knows it already.

Specifically, regarding a lower bound / comparison to existing works: It is currently not clear what the lower bound is for closeness testing with the $\mathcal{A}_k$ distance or for the continuous case of the privacy parameter regime. However, the paper mentioned in the related work -- of Acharya et al (2018) --- provides a lower bound for identity testing, which is a simpler task than closeness testing. The lower bound they present is $O\left(\frac{\sqrt{n}}{\alpha^2}+\frac{\sqrt{n}}{\alpha\sqrt{\epsilon}} + \frac{n^{1/3}}{\alpha^{4/3}\epsilon^{2/3}}  + \frac{1}{\alpha\epsilon} \right)$ (when $n$ is the size of the domain). Additionally, the paper of Diakonikolas et al (2015) proves that the lower bound for the non-private parameter is $O\left(\frac{\sqrt{k}}{\alpha^2}+\frac{k^{4/5}}{\alpha^{6/5}}\right)$. It can be concluded that the lower bound is at least $\left(\frac{\sqrt{k}}{\alpha^2}+\frac{k^{4/5}}{\alpha^{6/5}}+\frac{\sqrt{k}}{\alpha\sqrt{\epsilon}} + \frac{k^{1/3}}{\alpha^{4/3}\epsilon^{2/3}} + \frac{1}{\alpha\epsilon} \right)$. Our result is almost the same as the lower bound, except for the term $\frac{\sqrt{k}}{\alpha\epsilon}$ (up to a polylog factors). This term is the result of the fact that testing for $\mathcal{A}_k$ distance uses the $L_2$ norm based tester, resulting in the term $\frac{\sqrt{k}}{\alpha\epsilon}$. This discussion will make appear in the camera-ready version of the paper.

As we promise to edit this version as to include your comments, we believe that ultimately our result - especially due to its many applications detailed in Table 1 - merits publication in NeurIPS. We humbly hope that you agree.

Specific reviewers' comments are provided as well.

---

### Decision · Program_Chairs · 2024-09-25

**Decision:**

Accept (poster)

**Comment:**

This paper studies the sample complexity of the problem of equivalence testing for continuous distributions under approximate differential privacy, and presents a new algorithm in this setting. Specifically, it adapts the algorithm of Diakonikolas, Kane, and Nikishkin (FOCS 2015) to the differentially private setting.

The reviewers are in agreement that this paper should be accepted, with the main concern being around the writing of the paper and the fact that some definitions are omitted. The authors are expected to address this concern in the final version as they promised in the rebuttal.